# Scale-variance in the carbon dynamics of fragmented, mixed-use landscapes estimated using Model-Data Fusion

David T. Milodowski[1,2], T. Luke Smallman[1,2], and Mathew Williams[1,2]

[1]School of GeoSciences, University of Edinburgh, Edinburgh UK
[2]National Centre for Earth Observations, University of Edinburgh, UK

**Correspondence:** D. T. Milodowski (d.t.milodowski@ed.ac.uk)

**Abstract.** Many terrestrial landscapes are heterogeneous. Mixed land cover and land-use generate a complex mosaic of fragmented ecosystems at fine spatial resolutions with contrasting ecosystem stocks, traits and processes, each differently sensitive to environmental and human factors. Representing spatial complexity within terrestrial ecosystem models is a key challenge for understanding regional carbon dynamics, their sensitivity to environmental gradients, and their resilience in the face of climate
change. Heterogeneity underpins this challenge due to the trade-off between the fidelity of ecosystem representation within modelling frameworks and the computational capacity required for fine-scale model calibration and simulation. We directly address this challenge by quantifying the sensitivity of simulated carbon fluxes in a mixed-use landscape in the UK to the spatial resolution of the model analysis. We test two different approaches for combining EO data into the CARDAMOM Model-Data Fusion (MDF) framework, assimilating time series of satellite-based Earth Observation (EO) derived estimates of ecosystem
leaf area and biomass stocks to constrain estimates of model parameters and their uncertainty for an intermediate complexity model of the terrestrial C cycle. In the first approach, ecosystems are calibrated and simulated at pixel-level, representing a "community average" of the encompassed land cover and management. This represents our baseline approach. In the second, we stratify each pixel based on land-cover (e.g. coniferous forest, arable/pasture etc.), and calibrate the model independently using EO data specific to each stratum. We test the scale-dependence of these approaches for grid resolutions spanning $1°$
to $0.05°$ over a mixed land-use region of the UK. Our analyses indicate that spatial resolution matters for MDF. Under the "community-average" baseline approach biological C fluxes (GPP, $R_{eco}$) simulated by CARDAMOM are relatively insensitive to resolution. However, disturbance fluxes exhibit scale-variance that increases with greater landscape fragmentation, and for coarser model domains. In contrast, stratification of assimilated data based on fine-resolution land-use distributions resolved the resolution dependence, leading to disturbance fluxes that were 40-100% higher than the baseline experiments. The differ-
ences in simulated disturbance fluxes result in estimates of the terrestrial carbon balance that suggest a C sink in the stratified experiment that is weaker than the stratified experiment. We also find that stratifying the model domain based on land-use leads to differences in the retrieved parameters that reflect variations in ecosystem function between neighbouring areas of contrasting land-use. The emergent differences in model parameters between land-use strata give rise to divergent responses to future climate change. Accounting for fine-scale structure in heterogeneous landscapes (e.g. stratification) is therefore vital
for ensuring the ecological fidelity of large-scale MDF frameworks. The need for stratification arises because land-use places strong controls on the spatial distribution of carbon stocks and plant functional traits, and on the ecological processes control-

ling the fluxes of C through landscapes, particularly those related to management and disturbance. Given the importance of disturbance to global terrestrial C fluxes, together with the widespread increase in fragmentation of forest landscapes, these results carry broader significance for the application of MDF frameworks to constrain the terrestrial C-balance at regional and national scales.

## 1 Introduction

Over the past decade, terrestrial ecosystems have provided a global net carbon (C) sink sequestering $\sim 3.4\pm0.9$ PgC yr$^{-1}$, $\sim 30\%$ of anthropogenic $CO_2$ emissions, despite estimated emissions of $\sim 1.6\pm0.7$ PgC yr$^{-1}$ associated with land-use and land-cover change (Friedlingstein et al., 2020). The future trajectory of the terrestrial carbon sink will therefore have a significant impact on global efforts to achieve the goal of the UN Framework Convention on Climate Change to avoid dangerous climate change, reaffirmed in the Glasgow Climate Pact (UNFCCC, 2021). Quantification of spatial and temporal variations in exchange magnitude, alongside their associated uncertainties, are therefore essential to understanding the stability of the terrestrial carbon sink in the face of rapid environmental change (Hurlbert et al. , 2019), and prerequisite to robust national reporting of land-based $CO_2$ emissions and their attribution to different sectors (Grassi et al., 2017; Jones and Friedlingstein, 2020; McGlynn et al., 2022). Terrestrial biosphere models provide a means of quantifying the land carbon balance in a systemic, ecologically coherent way (Bonan et al., 2018). However, the current and future dynamics of terrestrial C exchange are highly uncertain, largely due to uncertainties in the structure and parameter constraints of the biosphere models themselves (Lovenduski and Bonan , 2017; Smallman et al., 2021).

Global land-use and land-cover change has increased the fragmentation of ecosystems, creating highly heterogeneous landscapes that host a mosaic of land-cover and uses (Lindenmayer and Fischer, 2013; Brink et al., 2017; Matricardi et al., 2020). This heterogeneity juxtaposes ecosystems with contrasting C stocks, traits and ecological processes, management, and environmental sensitivity, at length-scales of 10-100m. For example, within the UK, landscapes comprise a patchwork of managed arable land and pasture, semi-natural and plantation woodland, heath and settlements (Figure 1). Insight into the dynamics of this patchwork of ecosystems has been greatly accelerated by the proliferation of Earth Observation (EO) data from satellites that monitor ecosystems with ever-increasing spatial and temporal resolution (Exbrayat et al., 2019). A major challenge is to synthesise this expanding range of EO data to generate systemic understanding of the terrestrial C cycle, thus transforming ecosystem observation into ecological understanding that can inform policy development and facilitate land management (Smallman et al., 2022).

Model-Data Fusion (MDF) frameworks provide the means to integrate EO observations with spatially explicit process-based ecosystem models that encapsulate our understanding of how C flows through ecosystems (Luo et al., 2011), thereby providing key, mass-balanced, constraints on the fluxes of C between the atmosphere and land surface alongside their associated uncertainties (Niu et al., 2014; Bloom et al., 2016; Peylin et al., 2016; MacBean et al., 2018; Smallman et al., 2021). MDF frameworks that exploit intermediate complexity models of the terrestrial C cycle, such as CARDAMOM (Bloom et al., 2016; Exbrayat et al., 2018; Lopez-Blanco et al., 2019; Smallman et al., 2021), are able to generate "local" calibrations based on

pixel-level inversions of EO and auxiliary data streams. Calibrating ecosystem models to local data is important, because the functional traits of ecosystems vary in space (Smith et al., 2013; Reich et al., 2014; Butler et al., 2017; Exbrayat et al., 2018; Lopez-Blanco et al., 2019; Smallman et al., 2021), with trait differences within biomes often exceeding differences between biomes (Van Bodegom et al., 2012; Butler et al., 2017); failure to account for such variations may lead to biases in the estimated dynamics (Scheiter et al., 2013; Exbrayat et al., 2018). However, the computational intensity of large-scale MDF frameworks limits their spatial resolutions to 10-100 km, several orders of magnitude greater than the length scales relevant to differentiating the ecosystems within landscape mosaics (e.g. Kaminski et al., 2012; Smith et al., 2013; Kuppel et al., 2014; Bloom et al., 2016; Peylin et al., 2016; Yin et al., 2020; Smallman et al., 2021).

The scale disparity between model domains and the ecological fabric those domains represent poses a major challenge to large-scale modelling of terrestrial C dynamics in heterogeneous landscapes (Stoy et al., 2009; Fisher et al., 2020; Levy et al., 2022). In typical spatially distributed MDF applications, available observations are aggregated to pixel-level "community averages" prior to inversion. There are usually sufficient degrees of freedom in ecological process models to fit the observed temporal changes in aggregated stocks and fluxes, based on available observation constraints (Beven et al., 2006; Famiglietti et al., 2021). Nevertheless their ecological fidelity may be limited in heterogeneous landscapes, for which the parameters retrieved by these "community-average" models provide intermediate representations of the distinct ecosystems present. This limitation compromises efforts to attribute fluxes to specific land-uses and raises potential for significant sources of bias when estimating the terrestrial carbon balance and its environmental sensitivity. Firstly, the C-cycle represents the interplay of a number of nonlinear ecological processes, and therefore upscaling raises the familiar foe of Jensen's inequality (Jensen, 1906; Levy et al., 2022), whereby for a set of input variables, $X$, the expectation value for a nonlinear function $f$ (i.e. $E[f(X)]$), will not yield the same estimate as the same nonlinear function applied to the average values of those variables ($f(E[X])$), leading to scale-variance. In the case of terrestrial C fluxes, land-use places strong controls both on the distribution of carbon stocks and plant functional traits within the landscape, but also on the processes controlling the fluxes of C through landscapes, particularly those related to exogenous processes such as management and disturbance. The inversion of the pixel-average environmental signals may provide biased diagnostics, in particular where pixels comprise a mixture of different ecosystem types and management. As the model domains become coarser, the length-scales over which the environmental signal is averaged increases. Failing to account for the co-location of stocks and process imposed by land-use in mixed-use landscapes (e.g. concentration of C stocks in woodland, where timber harvest is focused) provides a clear source of potential systematic scale-variant bias in derived flux estimates across large scales. Additionally, "community average" models may miss or poorly represent processes specific to certain land-uses (White et al., 2019; Kondo et al., 2020). To ensure the ecological fidelity of large-scale ecosystem C-cycle models, it is therefore vital to adequately capture the essential processes controlling the fluxes of C through these different ecosystems, and their potentially divergent temporal dynamics and environmental sensitivities (Levy et al., 2022).

In this study, we specifically address the impact of the resolution trade-off in spatially explicit MDF frameworks between ecological fidelity and computational intensity by investigating how simulated carbon cycling in a mixed-use landscape in the UK respond to the spatial resolution of the model grid. We test two different MDF approaches that assimilate EO information of

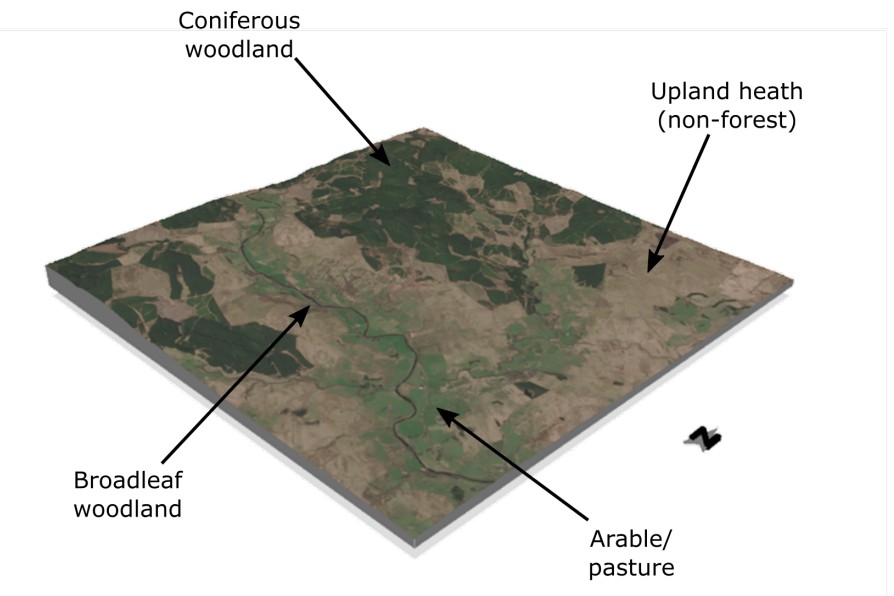

**Figure 1.** Perspective view of Sentinel 2 imagery over a typical landscape sampled from the study area illustrating the fine-scale mosaic of land-use characteristic of this region. The spatial extent of the displayed domain is 10 km x 10 km, and comprises part of the North River Tyne catchment in Northumberland, with the spatially extensive coniferous woodlands of Kielder Forest encroaching into the NW of the scene (top). Contains modified Copernicus Sentinel data [2021].

ecosystem characteristics to constrain model parameters and uncertainty for an intermediate complexity model of the terrestrial C cycle, DALEC (Williams et al., 2005; Bloom et al., 2016; Smallman et al., 2017, 2021). In the first approach, ecosystems are calibrated and modelled at the pixel level, representing a "community average" of the encompassed land-cover and management. This corresponds to the approach commonly employed in large-scale ecosystem MDF frameworks (e.g. Smith et al., 2013; Bloom et al., 2016; Yin et al., 2020; Smallman et al., 2021). In the second, we stratify each pixel based on land-cover,

and calibrate the model independently using remotely sensed data specific to each stratum, aligning more closely with the tiled Plant Function Type (PFT) approach employed in many terrestrial biosphere models (e.g. Sitch et al., 2008; Kaminski et al., 2012; Kuppel et al., 2014). The novelty of introducing stratification within a MDF context is that we use fine-scale ecosystem information contained within EO data to retrieve locally calibrated parameter ensembles for the ecosystems represented by each stratum, and therefore retain that key advantage of MDF systems, which enables calibrated traits to vary across environmental

gradients, within the constraints of the available observations and ecological knowledge (Smallman et al., 2022). However, by stratifying pixels we attempt to minimise the extent to which the environmental signals being inverted are averaged across functionally distinct ecosystems, thus ameliorating one source of scale-variant error in the estimated C fluxes. Furthermore, the model parameters retrieved through MDF synthesise the ecological information relayed by the assimilated data streams, within the constraints imposed by model structure and data quality. At coarser scales, aggregating observation streams results in the

loss of ecological information, which we expect to be particularly marked in heterogeneous landscapes. Stratification provides one mechanism through which this ecological information loss may be reduced. Of course, the ecological fidelity of the model calibrations may still be limited where the model structure cannot adequately represent important processes, or where there are systematic errors and biases in the assimilated data; stratification by itself does not resolve these components of ecological fidelity, but does open avenues through which they may be addressed.

We test the two MDF approaches - the novel sub-pixel stratification approach and the traditional pixel average (baseline) approach - on grid resolutions spanning $1°$ to $0.05°$. Specifically we address the following hypotheses:

- *H1*: estimated C fluxes will be scale variant, with stronger resolution-sensitivity exhibited by exogenous fluxes (i.e. disturbance) compared to biogenic fluxes (e.g. GPP).

- *H2*: C fluxes will be more consistent across grid resolutions when the framework explicitly accounts for sub-pixel heterogeneity in land-use; estimates from the baseline (unstratified) experiments will converge on the stratified estimates at finer spatial resolutions.

- *H3*: without accounting for fine-scale variations in land cover, the ecological information embedded within the retrieved parameters will be degraded when assimilating data streams at coarser resolutions, resulting in divergent carbon dynamics in simulations of future trajectories.

Using MDF to combine models and data at local scale offers huge potential for rigorous quantification of the state and dynamics of the terrestrial C cycle across large spatial scales, with propagation of uncertainty through analyses (Bloom et al., 2016; Smallman et al., 2022). In testing the hypotheses outlined above, we seek to address a key challenge relating to the mismatch between the scales of ecological processes and of large-scale MDF frameworks through the development of a novel stratified MDF framework. Our approach retains the core advantages of MDF, namely local calibration with local information, while also capturing the fine-scale ecosystem heterogeneity common to fragmented or mixed-use landscapes.

## 2 Methods

### 2.1 Study Area

The study area for this site covers northern England and the Scottish Borders, spanning approximately 30,000 km$^2$ across three degree of longitude and one degree of latitude (Figure 2). The region comprises a mosaic of land-cover types, including coniferous plantation forest (including the nationally significant forestry estates of Kielder Forest, Eskdalemuir Forest and Galloway Forest), fragments of broadleaf woodland, upland heath, arable agriculture and pasture. The longitudinal extent stretches from coast to coast, from the Firth of Clyde in the West to the North Sea in the East. Elevation varies from sea level to a high of 978 m on Scafell Pike in the Lake District. These gradients in longitude and elevation are associated with gradients in both precipitation and temperature (Jenkins et al., 2009). Precipitation decreases from west to east in response to the prevailing

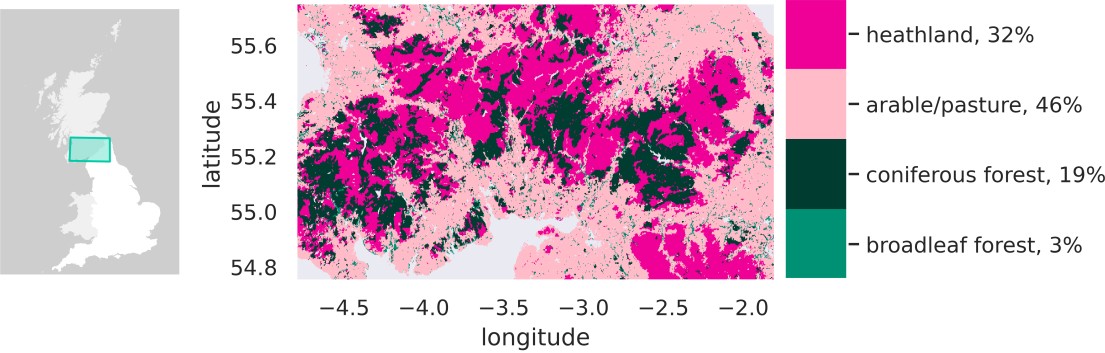

**Figure 2.** Map of the study area, spanning one° of latitude and three° of longitude across southern Scotland and northern England. The land-cover types displayed are aggregated from the LCM2015 land-cover map of Great Britain (Rowland et al., 2017), regridded to approximately 300m resolution.

westerly wind direction and orographic enhancement of rainfall in areas of high topography; temperature gradients are broadly controlled by elevation.

## 2.2 Model-data fusion with CARDAMOM (CARbon DAta MOdel fraMework)

### 2.2.1 DALEC

At the core of our model-data fusion framework sits DALEC, an intermediate complexity model of the terrestrial C cycle (Williams et al., 2005; Bloom and Williams, 2015; Smallman et al., 2017; Famiglietti et al., 2021). DALEC is a mass balance model of the C cycle with carbon moving through different pools based on parameterised fluxes (Figure 3). A number of variants of DALEC have been created representing ecosystem carbon dynamics with varying degrees of complexity (Famiglietti et al., 2021; Smallman et al., 2021). The specific version of DALEC used here corresponds to the C6 model outlined in Famiglietti et al. (2021) which combines the C-cycle structure from Bloom and Williams (2015) with the revised photosynthesis model from Smallman and Williams (2019). There are four live biomass pools, specifically relating to carbon stored in foliage, labile carbon, fine roots and wood, and two dead organic carbon pools: litter and soil organic carbon. Carbon enters the system through GPP, modelled using the photosynthesis model ACM2 (Smallman and Williams, 2019), wherein GPP is simulated as a function of modelled leaf area, estimated canopy photosynthetic efficiency, absorbed solar radiation, atmospheric $CO_2$ concentration, air temperature and a stomatal conductance model that balances potential water supply from the soil (assumed to be at field capacity) through the roots with atmospheric demand, determined by absorbed solar radiation and VPD. Carbon is lost from the system via autotrophic and heterotrophic respiration. NPP is allocated between autotrophic respiration and the live pools based on fixed fractions. Canopy growth is driven by a combination of direct allocation from GPP and transfer of carbon from the labile pool. The flux of carbon from the labile pool to foliage and canopy senescence, driving litter-fall, are controlled by a simple day-of-year phenology model with a parameterised leaf life span (Bloom and Williams, 2015). Carbon

flows from the roots and wood into the litter and soil organic carbon pools respectively based on first order turnover rates. Heterotrophic respiration fluxes also follow first order kinetics, but with an exponential temperature sensitivity. A full list of model parameters is provided in the appendix (Table A1). The relative simplicity compared to other terrestrial biosphere models make DALEC amenable to calibration in model-data fusion frameworks, and allows propagation of uncertainties through large ensemble simulations (Bloom et al., 2016; Exbrayat et al., 2018; Famiglietti et al., 2021; Smallman et al., 2021).

### 2.2.2   Model-Data Fusion

Our model-data fusion framework, CARDAMOM (Bloom et al., 2016), uses a Bayesian approach within an Adaptive Proposal Markov Chain Monte Carlo (AP-MCMC) framework (Haario et al., 2001) that can assimilate a range of information (Bloom et al., 2016), including remotely sensed LAI and aboveground biomass (see Section 2.3). The premise of the approach is to take driving data describing the meteorology and disturbances such as forest clearance and fire, and search the model parameter space to find parameter combinations that provide simulated dynamics that are consistent with the available data. Specifically, given a set of observations, $O$, with uncertainty $\sigma$, the probability of a given parameter set $x$, $P(x|O)$, is calculated as a function of the likelihood of the observations given the current parameters, $P(O|x)$, and any prior knowledge on the parameter distributions, $P(x)$:

$$P(x|O) \propto P(O|x) \cdot P(x) \tag{1}$$

The likelihood $P(O|x)$ is calculated based on the misfit between the $N$ available observations and the equivalent simulated state variables and fluxes for each parameter set, $M$:

$$P(O|x) = exp\left(-0.5 \cdot \sum_{n=1}^{N}\left(\frac{O_n - M_n}{\sigma_n}\right)^2\right) \tag{2}$$

We use the Gelman-Rubin's convergence criterion to determine whether multiple chains at each pixel have converged. The AP-MCMC (Haario et al., 2001) does not stipulate or target an acceptance rate; the emergent acceptance rate typically varies between 5 and 25 %. The covariance matrix used in adapting the parameter sampling is generated from an initial phase of the MCMC. No hyperparameters are estimated as part of the process.

To facilitate the calibration process, we employ a series of Ecological Dynamic Constraints, EDCs (Bloom and Williams, 2015; Smallman et al., 2017). EDCs comprise a series of mathematical rules and functions that impose conditions on the inter-relationships between model parameters to ensure ecological "realism" in the accepted parameter sets, and are based in ecological theory (Bloom and Williams, 2015). For example, the turnover of the wood carbon pool must be slower than foliage turnover. Where EDCs are not satisfied, the likelihood is set to zero. By restricting the acceptable parameter space, the EDCs therefore reduce the effective model complexity (Famiglietti et al., 2021), and tend to reduce bias and equifinality in the calibrated ensembles (Bloom and Williams, 2015). The resulting ensemble of parameter sets encapsulate the uncertainty in the calibration within the available observational constraints.

### 2.2.3 Stratification approach for mosaic landscapes

Our approach to handling fine-scale heterogeneity during the model-data fusion process is based on sub-pixel stratification based on land-use (Figure 3). Stratification is achieved by sampling the spatially gridded EO data products at their native spatial resolution based on a reference land-cover map, resampled to the same resolution using the modal category.

The specific land-cover product used is the LCM2015 land-cover map produced by the UK Centre for Ecology and Hydrology (CEH) (Rowland et al., 2017), which we aggregate to four classes: coniferous woodland, broadleaf woodland, arable/pasture and heathland, which includes semi-natural grasslands and widespread areas of non-wooded upland heath (Figure A1). Urban and coastal areas are masked from all analyses. For each pixel, separate ensembles are calibrated independently, yielding a suite of ensembles that maintain the ecological fidelity of the calibrated parameters. While aggregation of observations to coarser resolution unavoidably results in information loss, stratification preserves the distinctions between functionally distinct ecosystems, and therefore in this sense, the ecological fidelity of the resultant suite of ensembles is maintained within the limitations of the model structure and data quality. This is a contrast to the "traditional" model-data fusion approach, which aggregates the data constraints into pixel-level "community-averaged" prior to calibration, yielding calibrated parameter combinations that may be attempting to account for a multitude of distinct ecological processes. However, when considering the ecological fidelity of the calibrated models, it is important to note that CARDAMOM will attempt to find parameters that satisfy the observation constraints and their uncertainties, and therefore systematic errors associated with particular data streams will propagate to lead to parameter estimates that do not provide a good representation of the ecology. We discuss this in relation to the impact of overly seasonal LAI observations for conifer woodland canopies and the resultant impact on canopy parameters in section 4.3.

The stratification approach is very flexible. The number of categories can be refined as necessary, within the data constraints. Regarding aggregation of uncertainties, we do not have constraints on the extent to which pixel uncertainties are correlated in space. Therefore for each stratum, spatial aggregation of uncertainty conservatively assumes correlated uncertainties (Exbrayat et al., 2018). However, we assume that individual strata, representing different ecosystems, are uncorrelated with other strata when aggregating sub-pixel ensembles to pixel-level. For simplicity of comparison across the experiments in this study against the baseline experiments (i.e. no stratification), we use only one model structure across all strata, and pre-process the assimilated data streams in the same way. For strata where woody tissues are not part of the dominant vegetation types, for example in areas covered by crop and pasture, the $C_{Wood}$ pool also provides a reservoir for non-woody structural tissue, with the differential allocation patterns and turnover rates reflected in the retrieved parameters. Importantly, different ecosystems could in the future be modelled with distinct, ecosystem-specific models that better capture their functional process dynamics. Relevant ecosystem-specific model variants have previously been integrated within the CARDAMOM framework, for example woodlands (Smallman et al., 2017), pasture (Myrgiotis et al., 2021) and arable agriculture (Revill et al., 2021). Given the computational limitations on the resolution of the model domain, stratification would be prerequisite to the inclusion of ecosystem-specific models within regional CARDAMOM applications.

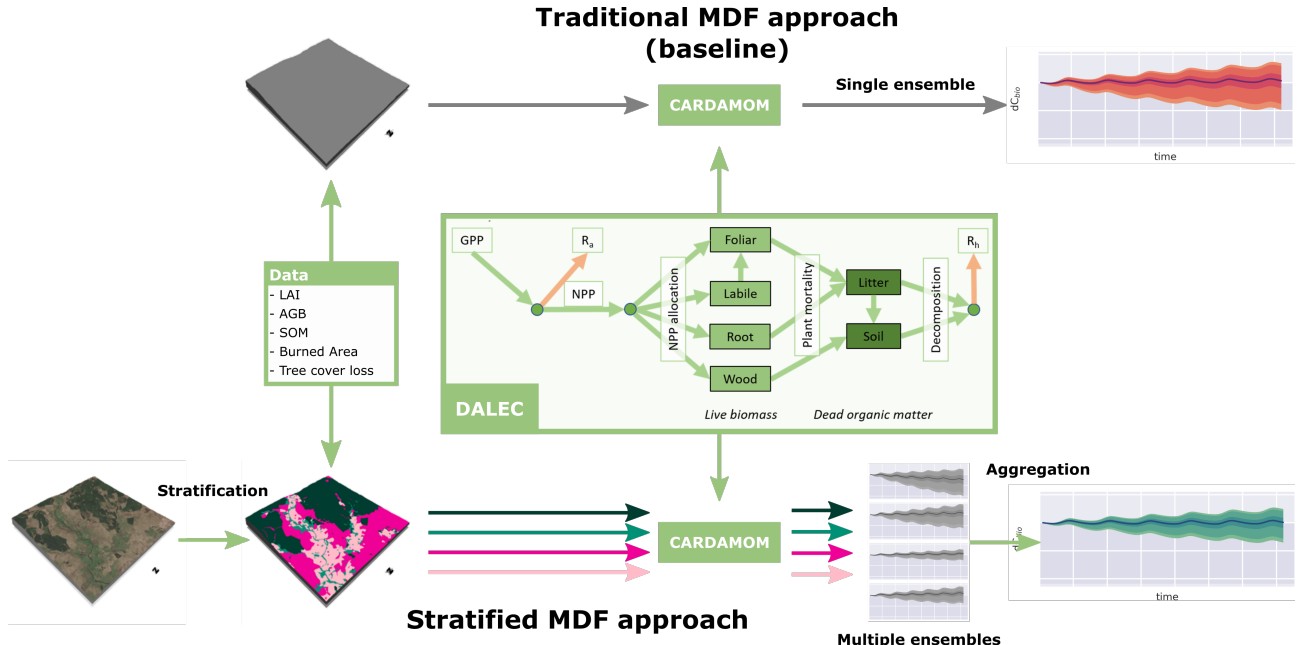

**Figure 3.** Schematic flow diagrams illustrating the different model-data fusion approaches employed in this study: the "traditional" model-data fusion approach whereby the input data are aggregated to pixel-level "community averages" and stratification based on the land-use leading to calibration of a suite of land-use specific ensembles. At the heart of both MDF approaches sits DALEC, an intermediate complexity model of the terrestrial C cycle (Bloom and Williams, 2015). While the presented time series show specifically the change in live biomass, $dC_{bio}$, it is important to note that similar information is retrieved for all fluxes and stocks within DALEC, alongside pixel-specific parameter ensembles.

## 2.3 Data

### 2.3.1 Meteorological drivers

Meteorological drivers, comprising temperature, shortwave radiation, vapour pressure deficit (VPD) and wind speed, are drawn from the CRU-JRAv1.1 dataset, a 6-hourly $0.5 \times 0.5°$ reanalysis (CRU, 2019). Atmospheric $CO_2$ concentration is taken from the Mauna Loa global $CO_2$ concentration (www.esrl.noaa.gov/gmd/ccgg/trends/, accessed: 22/08/2020).

### 2.3.2 Copernicus LAI 300 m

    LAI data is obtained from the 300 m Copernicus LAI product v1.0 (Fuster et al., 2020) for period 2014-2019. The LAI estimates
in the Copernicus 300 m product represent 10-day composites from daily estimates of LAI that are generated from to daily Top-of-Atmosphere input reflectances detected by the PROBA-V satellite by applying a neural network. These 10-day LAI estimates were aggregated to monthly averages prior to assimilation. Pixel-wise uncertainty estimates are also provided with this product, calculated as the root-mean-square difference between the individual daily neural network estimates and the 10-

day average. Previous work has indicated that these uncertainty estimates underestimate the true uncertainties associated with this product (Zhao et al., 2020). We therefore used a more conservative temporal aggregation approach based on the maximum uncertainty within the aggregation period.

### 2.3.3 ESA Biomass CCI Aboveground Biomass 2017, 2018

Aboveground biomass (AGB) estimates and associated uncertainty were extracted the global maps published within the ESA Biomass CCI collection (Version 2), comprising two estimates for the years 2017 and 2018 with as spatial resolution of 100 m (Santoro et al., 2021). The ESA CCI Biomass data are derived from Synthetic Aperture Radar (SAR) backscatter data, specifically ALOS PALSAR L-band SAR backscatter combined with Sentinel-1 C-band SAR backscatter. Uncertainty estimates are provided with this product, calculated as the standard deviation associated with the AGB estimate after propagating errors through the SAR measurement, SAR-AGB modelling framework and merging of L-band and C-band estimates into an overall AGB estimate (Santoro et al., 2021).

The DALEC wood carbon pool represents the combination of above- and below-ground carbon (i.e. including the coarse root component). The contribution from below-ground biomass (BGB) to the woody biomass pool, alongside the associated uncertainty, is modelled using an allometric relationship following Saatchi et al. (2011):

$$BGB = 0.489 \cdot AGB^{0.89} \tag{3}$$

### 2.3.4 SoilGrids2 Soil Organic Carbon (SOC)

Soil organic carbon estimates and associated uncertainties were obtained from SoilGrids2, which provide 250m resolution spatial maps of depth profiles for various soil properties (Poggio et al., 2021). These maps were produced using EO and auxiliary spatial data within a machine learning framework trained on over 230,000 individual soil profile observations. The extracted SOC estimates are used to set a prior constraint on the initial SOC stock. As there is no date associated with the SoilGrids2 dataset we use these estimates, with their uncertainty, to provide a prior constraint on the initial SOC stocks. This contrasts with our treatment of the LAI and AGB data, which are associated with specific time periods and therefore used as observational constraints on the simulated time series. However, the aggregation of the original SoilGrids2 data layers for the baseline and stratified experiments follows the same procedure.

### 2.3.5 Disturbance

Disturbance is imposed on DALEC based on satellite observations of tree cover loss and burned area. Disturbances related to tree cover loss are driven by observations from the Global Forest Watch (GFW) dataset (Hansen et al., 2013), which provides annual constraints on tree cover loss at 30 m resolution based on Landsat data. Note that other mechanisms of disturbance, such as agricultural harvests and pasture management, are not considered in the current analysis. To convert area estimates of tree cover loss into changes in C stocks, we use a simple clearance model in which a fraction of the C stored in $C_{wood}$, $C_{foliage}$ and $C_{labile}$ is removed based on the pixel fraction (or stratum-specific sub-pixel fraction) identified within the GFW dataset as

experiencing tree cover loss. In practise, most tree cover loss occurs in the conifer woodlands, and is therefore concentrated
in these woodlands in the stratified analysis, compared to the baseline experiments, in which we do not consider the sub-pixel
distribution of land cover. Fire is imposed based on monthly aggregated burnt area fractions in the MODIS MCD64A1 product
(Giglio et al., 2018), which maps fire-affected areas at 500m resolution based on changes in surface reflectance, although
the occurrence of MODIS-detected fires throughout the model domain was very low. Emissions from fire are estimated by
assuming a fraction of simulated biomass either undergoes combustion, therefore immediately released to the atmosphere, or
is transferred to the litter pool, based on tissue specific combustion-completeness factors (Exbrayat et al., 2018).

## 2.4  Experimental setup

To test how simulated C fluxes varied with grid resolution we calibrated DALEC across the target domain at four different
grid resolutions: $0.05°$, $0.25°$, $0.50°$ and $1.00°$, at a monthly time-step spanning the period 2014-2019. We compared the
retrieved parameters and simulated C fluxes for two MDF approaches: the proposed stratified CARDAMOM calibration that
explicitly accounts for sub-pixel heterogeneity in land-use, and the traditional pixel aggregate CARDAMOM calibration. The
latter serves as a baseline. In the baseline experiments, the observation streams were aggregated to the domain resolutions,
and these "community-average" environmental signals were assimilated into a single ensemble. In the stratified experiments,
the individual observation streams were stratified at their native resolutions based on the dominant category from the high-
resolution land cover map, and then these strata-specific subsets of observations were aggregated to the resolution of the model
domains before assimilation. In all cases we use the same underlying DALEC model structure within the MDF framework
(Figure 3). Emergent differences in the retrieved parameters, stocks and fluxes between experimental runs are therefore a
consequence of the resolution at the observations are aggregated, rather than ecosystem-specific differences in model structure.
The resolution of the meteorological data in all cases is $0.5°$, therefore our analysis does not allow us to test the extent to which
resolving fine-scale variations in meteorological forcing impacts on the overall C balance.

We characterise the calibration performance for each ensemble based on the RMSE and the bias with respect to the $N$
assimilated observations:

$$\text{RMSE} = \sqrt{\frac{\sum_{n=1}^{N}(O_n - M_n)^2}{N}} \tag{4}$$

$$\text{Bias} = \frac{\sum_{n=1}^{N}(O_n - M_n)}{N} \tag{5}$$

In both cases, we calculate pixel-level metrics, and then weight the contributions from individual pixels when aggregating
across the domain based on the fractional coverage contributed by each stratum. As the observations are also associated with
significant uncertainty, we also consider the ratio of the RMSE and Bias to the product uncertainty as a measure of agreement
within the uncertainty constraints provided by the assimilated data. Values $>1$ would indicate situations where the model was

not able to fit the observations to within their associated uncertainty. Different data streams may provide inconsistent and/or in-compatible information, for example due to data biases or incorrect specification of uncertainty citepzhao2020. This could lead to larger RMSE/$\sigma$ ratios as CARDAMOM attempts to balance inconsistent information. Larger model-data mismatch could also indicate model structural error. Values <1 may indicate improved constraints based on the combination of assimilating complementary data streams and the ecological knowledge embedded in the model and EDCs.

We are able to address our first two hypotheses (*H1*, *H2*), relating to the impact of resolution and sub-pixel stratification on diagnostic analyses of C cycle dynamics, by comparing the changes in C stocks and fluxes over the data assimilation period. *H3* is addressed by comparing the retrieved parameters for each run, including the individual land-use classes in the stratified analysis, and how these distributions shift depending on the land-use and on the spatial resolution of the analysis. To understand the potential impact of any emergent differences in the retrieved parameters on future trajectories, we then ran forward simulations of our DALEC ensembles to 2100 under the SSP2-4.5W m$^{-2}$ scenario extracted from the UK Earth System Model (UKESM; Sellar et al., 2019) contribution to CMIP6 (Eyring et al., 2016), which corresponds to a middle-of-the-road scenario with a projected mean global warming of 2.7$^o$C (O'Neill et al., 2016). We do not impose future disturbance fluxes, so emergent differences in C dynamics will be driven by the interactions between climate and the retrieved parameters for each ensemble. To avoid step-changes in meteorology between the historical meteorology (from observations) and future meteorology (simulated by UKESM) we apply the future trajectories for each scenario based on the anomaly in the UKESM forecast relative to 2019 (following Smallman et al., 2021).

## 3 Results

### 3.1 Calibration performance

The two MDF approaches tested provided comparable fits to the calibration data. Both the baseline and stratified CARDAMOM calibrations were able to fit the assimilated C$_{Wood}$ and LAI observations to well within the levels of observation uncertainty, with the RMSE between simulated and observed variables less than 50% of the uncertainties attached to the assimilated observations (Table 1, Figures A2, A3). In general the RMSE values were comparable between the stratified and baseline experiments for LAI (mean RMSE for C$_{Wood}$ across spatial resolutions: 14.0% for the baseline experiment; 13.6% for the stratified experiment), while the RMSE for C$_{Wood}$ was slightly lower for the stratified experiment (mean RMSE for C$_{Wood}$ across spatial resolutions: 14.0% for the baseline experiment; 13.6% for the stratified experiment). For both LAI and C$_{Wood}$, the RMSE tended to increase at finer spatial resolutions in both the baseline experiments and the aggregated stratified experiment (Table 1), although the resolution-dependent trend was not consistent between individual strata, Table A2. An increase in RMSE at finer spatial resolutions can be rationalised by the smoothing effect of aggregating the remote sensing products over larger spatial scales. This not only removes the impact of high frequency random noise in the assimilated signal, but also removes variability generated by local processes (e.g. management) that are not accounted for in the relatively simple treatment of canopy dynamics encoded in the model. In the stratified experiment, the bias in C$_{Wood}$ was dominated by the contributions from the woodland strata (Table A2), corresponding to their much greater C$_{Wood}$ stocks, which were over four times higher

**Table 1.** Summary of calibration performance, aggregated across the domains for the baseline and stratified experiments. $\sigma$ represents the uncertainty of the assimilated observation data, thus RMSE / $\sigma$ provides the ratio of the RMSE to the uncertainty attached to the observation constraint. For an equivalent breakdown of calibration performance of the individual strata in the stratified experiment, see Table A2. The values in parentheses following the RMSE and bias estimates indicate the percentage relative to the mean of the observations across the domain.

| Variable | Version | Metric | 1.00° | 0.50° | 0.25° | 0.05° |
|----------|---------|--------|-------|-------|-------|-------|
| $C_{Wood}$ | Baseline | RMSE / gCm$^{-2}$ | 215 (12.0 %) | 247 (13.9 %) | 342 (19.3 %) | 446 (25.3 %) |
| $C_{Wood}$ | Baseline | RMSE / $\sigma$ | 0.14 | 0.18 | 0.27 | 0.37 |
| $C_{Wood}$ | Baseline | Bias / gCm$^{-2}$ | 166 (9.3 %) | 182 (10.2 %) | 122 (6.9 %) | 67 (3.8 %) |
| $C_{Wood}$ | Baseline | Bias / $\sigma$ | 0.11 | 0.14 | 0.17 | 0.23 |
| $C_{Wood}$ | Baseline | Median gCm$^{-2}$ | 1925 | 1934 | 1873 | 1817 |
| $C_{Wood}$ | Stratified | RMSE / gCm$^{-2}$ | 140 (7.8 %) | 220 (12.4 %) | 291 (16.4 %) | 364 (20.7 %) |
| $C_{Wood}$ | Stratified | RMSE / $\sigma$ | 0.09 | 0.14 | 0.20 | 0.27 |
| $C_{Wood}$ | Stratified | Bias / gCm$^{-2}$ | -99 (-5.6 %) | -98 (-5.5 %) | -106 (-6.0 %) | -106 (-6.0 %) |
| $C_{Wood}$ | Stratified | Bias / $\sigma$ | -0.06 | -0.04 | 0.00 | 0.08 |
| $C_{Wood}$ | Stratified | Median gCm$^{-2}$ | 1687 | 1673 | 1666 | 1657 |
| LAI | Baseline | RMSE / m$^2$m$^{-2}$ | 0.36 (15.8 %) | 0.37 (16.2 %) | 0.40 (17.1 %) | 0.45 (19.4 %) |
| LAI | Baseline | RMSE / $\sigma$ | 0.35 | 0.36 | 0.38 | 0.43 |
| LAI | Baseline | Bias / m$^2$m$^{-2}$ | -0.02 (-0.8 %) | -0.02 (-0.9 %) | -0.02 (-1.0 %) | -0.03 (-1.1 %) |
| LAI | Baseline | Bias / $\sigma$ | -0.02 | -0.02 | -0.02 | -0.02 |
| LAI | Baseline | Median m$^2$m$^{-2}$ | 2.19 | 2.18 | 2.18 | 2.19 |
| LAI | Stratified | RMSE / m$^2$m$^{-2}$ | 0.35 (15.0 %) | 0.37 (16.6 %) | 0.40 (17.1 %) | 0.46 (19.6 %) |
| LAI | Stratified | RMSE / $\sigma$ | 0.33 | 0.35 | 0.38 | 0.43 |
| LAI | Stratified | Bias / m$^2$m$^{-2}$ | -0.03 (-1.3 %) | -0.02 (-0.8 %) | -0.03 (-1.1 %) | -0.04 (-1.6 %) |
| LAI | Stratified | Bias / $\sigma$ | -0.03 | -0.02 | -0.02 | -0.03 |
| LAI | Stratified | Median m$^2$m$^{-2}$ | 2.18 | 2.18 | 2.18 | 2.18 |

in coniferous woodland than arable/pasture, and six times higher than in the heathland class (Table A2, Figure A3). Notably, all strata, including the coniferous woodland class, contained a strong seasonal cycle of monthly LAI in both the assimilated observations and the simulations (Figure A2).

## 3.2 Terrestrial C budget and impact of spatial resolution on C flux estimates

Both the baseline and stratified CARDAMOM analyses estimated the net ecosystem exchange of C to be most likely a net sink of C over the calibration period. C uptake from the atmosphere via GPP was ~5 gC m$^{-2}$d$^{-1}$ (Table 2; for stratified experiment, 0.05°, GPP = 4.87 (3.75 - 5.95) gC m$^{-2}$d$^{-1}$) while C returned to the atmosphere via autotrophic and heterotrophic respiration (R$_{eco}$) was ~4.5 gC m$^{-2}$d$^{-1}$ (Table 2; for stratified experiment, 0.05°, R$_{eco}$ = 4.57 (3.33 - 6.07) gC m$^{-2}$d$^{-1}$). At increasingly fine spatial resolutions the model reveals greater spatial variability, reflecting the impact of landscape features and topography that are only adequately resolved at the finest grid scale (Figure A4). However, median estimates of GPP and R$_{eco}$ were relatively insensitive to the spatial resolution once aggregated across the spatial domain (Figure A5), varying by ≤0.05

**Table 2.** Summary of domain-aggregated carbon budgets for the baseline and stratified experiments. Fluxes are gross primary productivity (GPP), total ecosystem respiration (R$_{eco}$), cumulative changes in live (dC$_{bio}$) and dead (dC$_{soil}$) organic C pools integrated over the six year assimilation period (2014-2019), and carbon losses due to harvest and other tree cover loss (harvest). Values represent the median pixel level estimates averaged across the domain, alongside the 5% and 95% percentiles, i.e. assuming fully correlated uncertainties.

| Flux | Version | 1.00° | 0.50° | 0.25° | 0.05° |
|---|---|---|---|---|---|
| GPP / gCm$^{-2}$d$^{-1}$ | baseline | 4.99 (3.63 - 6.25) | 4.98 (3.65 - 6.27) | 4.98 (3.59 - 6.28) | 4.94 (3.55 - 6.26) |
| GPP / gCm$^{-2}$d$^{-1}$ | stratified | 4.88 (4.06 - 5.71) | 4.89 (3.98 - 5.75) | 4.91 (3.93 - 5.85) | 4.87 (3.75 - 5.95) |
| R$_{eco}$ / gCm$^{-2}$d$^{-1}$ | baseline | 4.51 (3.08 - 6.34) | 4.52 (3.07 - 6.41) | 4.54 (3.04 - 6.42) | 4.56 (3.06 - 6.47) |
| R$_{eco}$ / gCm$^{-2}$d$^{-1}$ | stratified | 4.52 (3.59 - 5.58) | 4.52 (3.52 - 5.68) | 4.53 (3.45 - 5.80) | 4.57 (3.33 - 6.07) |
| cumulative dC$_{bio}$ / gCm2 | baseline | 292.7 (-1021.6 - 1314.3) | 268.1 (-1090.7 - 1280.7) | 261.4 (-1061.4 - 1197.4) | 212.3 (-1098.4 - 1099.5) |
| cumulative dC$_{bio}$ / gCm2 | stratified | 127.6 (-474.5 - 527.4) | 126.6 (-523.7 - 572.5) | 126.6 (-556.1 - 622.9) | 128.6 (-730.3 - 724.1) |
| cumulative dC$_{soil}$ / gCm2 | baseline | 475.4 (-1565.1 - 2287.7) | 429.4 (-1675.8 - 2195.3) | 393.5 (-1684.3 - 2112.8) | 397.7 (-1610.9 - 2076.5) |
| cumulative dC$_{soil}$ / gCm2 | stratified | 257.9 (-778.7 - 1204.6) | 274.3 (-845.7 - 1268.1) | 296.5 (-926.1 - 1370.6) | 335.0 (-1112.3 - 1588.7) |
| harvest / gCm$^{-2}$d$^{-1}$ | baseline | 0.027 (0.013 - 0.047) | 0.029 (0.014 - 0.050) | 0.031 (0.014 - 0.057) | 0.038 (0.016 - 0.073) |
| harvest / gCm$^{-2}$d$^{-1}$ | stratified | 0.054 (0.020 - 0.112) | 0.053 (0.019 - 0.111) | 0.053 (0.019 - 0.110) | 0.054 (0.019 - 0.110) |

gC m$^{-2}$d$^{-1}$ (Table 2). The simulated uncertainties were smaller in the stratified experiment compared to the baseline for both GPP and R$_{eco}$ (Figure 7). The reduced uncertainty with stratification is a result of assuming independence between strata. If the uncertainty in these fluxes is assumed fully correlated across strata, then the combined uncertainty in the stratified experiment is comparable to that of the baseline experiment. The degree to which stratification reduces uncertainty is therefore determined by the extent to which strata are considered independent.

In contrast to GPP and R$_{eco}$, the disturbance flux exhibited a stronger resolution dependence in the baseline experiment (1.00°: 0.027 (0.013 - 0.047) gC m$^{-2}$d$^{-1}$; 0.05°: 0.038 (0.016 - 0.073) gC m$^{-2}$d$^{-1}$), while the disturbance flux is insensitive to resolution in the stratified experiment (0.054 (0.019 - 0.112) gC m$^{-2}$d$^{-1}$; see Table 2). In this set of experiments, the disturbance flux is driven by forest harvest (i.e. tree cover loss), as fire was negligible, affecting only sixteen pixels across the finest resolution domain across the entire period of analysis. The baseline experiments simulate a tendency towards increasing C$_{bio}$ stocks for all four spatial resolutions, with the median simulated sink strength increasing at coarser grid resolutions. This emergent scale-dependent sensitivity of dC$_{bio}$ is not shared by the stratified experiment, for which median dC$_{bio}$ is more consistent across the range of spatial resolutions. The disagreement in C accumulation between the two approaches is reduced at finer resolution model domains. Considering the median cumulative dC$_{bio}$, the resolution-dependent bias for the baseline experiment compared to the stratified experiment declines by ~50% moving from the 1.00° to the 0.05° domain (Table 2). However, the two approaches do not reach convergence even at 0.05° resolution (Figure A5).

Harvest fluxes in the stratified ensemble setup were between 2 and 1.4 times higher than the baseline ensemble, with the difference increasing systematically at coarser grid resolutions (Figure 5). Areas declining in dC$_{bio}$ are generally focused around the primary commercial forestry regions, where timber harvest is most abundant, for both the baseline and stratified experiments (Figure 6). Across these regions hosting significant areas of conifer woodland, the magnitudes of simulated net losses in the live carbon pools are generally greater for the stratified ensemble. Comparing the differences in dC$_{bio}$ estimated

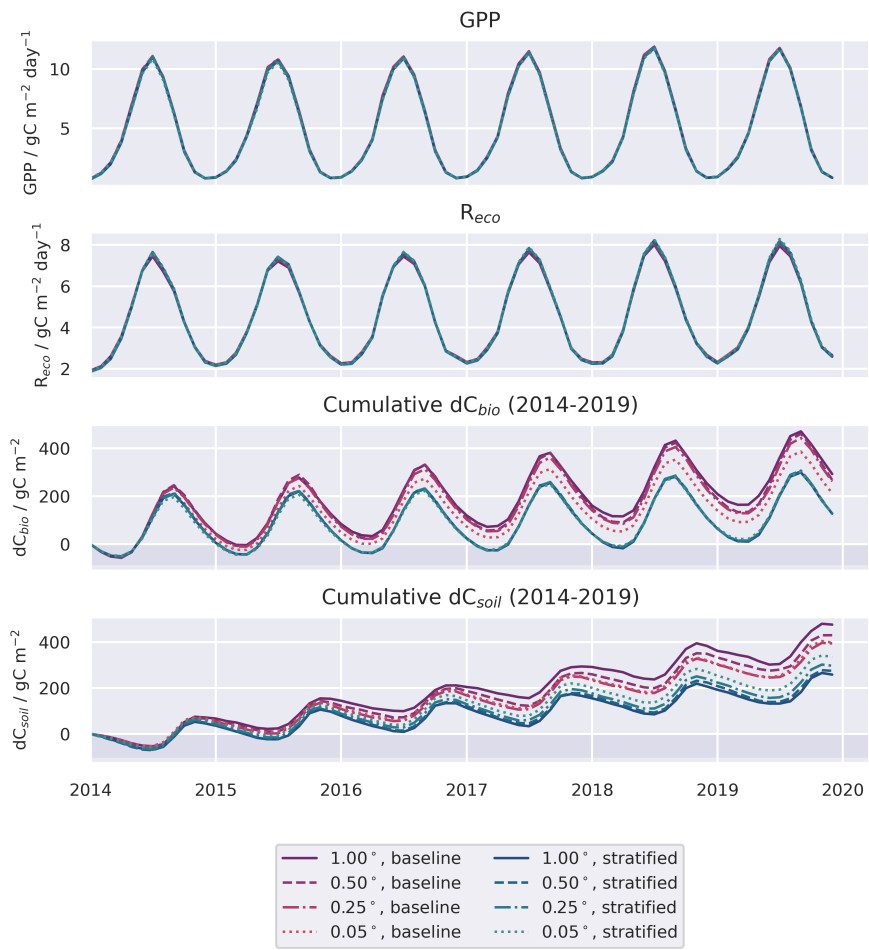

**Figure 4.** Spatially aggregated time series for GPP and ecosystem respiration ($R_{eco} = R_a + R_{het}$),and the cumulative change in carbon stocks in the live ($dC_{bio}$) and soil ($dC_{soil}$) pools, shown for the baseline and stratified ensembles for four spatial resolution domains. Only the median estimates are shown for clarity. Confidence levels for the $1°$ and $0.05°$ domains for the same time series are provided in Figure S2.

by the stratified and baseline approaches on a pixel-wise basis (Figure 6, A6), it is apparent that the stronger relative biases in $dC_{bio}$ correspond to parts of the domain with higher harvest fluxes. In turn, differences in harvest flux between the stratified and baseline experiments are strongly influenced by the level of sub-pixel heterogeneity (Figure 6, A6), with the difference in the harvest flux between the two experiments declining as the fraction of pixels covered by the coniferous woodland class (where both harvest and live C stocks are concentrated) approaches full coverage. In other words, there is little difference between the two experiments where pixels have homogeneous land-use.

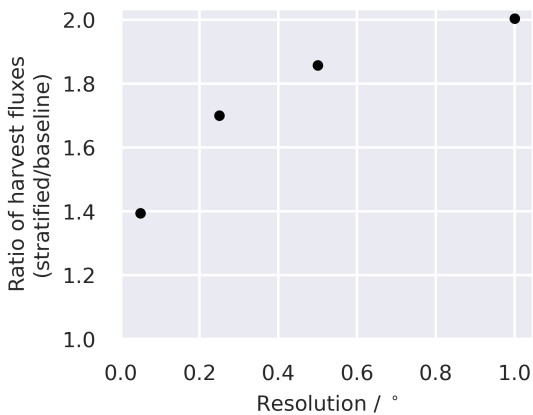

**Figure 5.** Ratio of the temporal mean harvest flux simulated in the stratified and baseline experiments, i.e (stratified/baseline), as a function of domain resolution. Points represent the median estimate aggregated across the spatial domain. Values >1 indicate baseline fluxes that are lower than the stratified case.

### 3.3  Impact of stratification on calibrated parameters and future C dynamics

The stratified data assimilation scheme reveals emergent differences between ecosystems, while traits retrieved for the baseline experiments characterised intermediate values (Figures 8, A7, A8, A9). In the baseline experiments, comparing across domain resolutions, it is apparent that aggregation to coarser spatial resolutions reduces the range of retrieved traits. The reduction in the widths of the retrieved parameter distributions highlights the loss of information relating to variations of fundamental aspects of ecosystem function in the baseline experiments, as the resolution of ecological gradients is lost. In contrast, it is evident that stratification leads to a reduction in the ecological information loss when aggregating to coarser resolutions, as the widths of the aggregated parameter distributions is maintained. The most pronounced differences in the calibrated parameters are associated with parameters closely related to the dynamics of the $C_{Wood}$ pool (Figure 8). Within the study landscape, live C stocks are concentrated in the woodland classes, particularly the conifer woodlands. Higher C stocks are reflected in greater woody productivity (Figure 8), higher residence times for the wood pool ($MRT_{Wood}$; Figure A7) and higher allocation fractions of NPP to the woody pool (Figure A9). In the baseline experiment, it is notable that large $MRT_{Wood}$ estimates are not represented within the distribution of median parameter estimates; this effect is magnified in the coarser domains, where the range of the simulated $MRT_{Wood}$ distribution contracts (Figures 8). In contrast, the longer-lived $C_{Wood}$ pools in the woodland classes are retained across the different resolution experiments in the stratified case (e.g. Figure 8). The longest soil C residence times ($MRT_{Soil}$ were retrieved for the heathland class (Figure A8), which includes upland areas underlain by peat deposits (Tanneberger et al., 2017). Again, the longer $MRT_{Soil}$ estimates of the heathland stratum are not well represented within the baseline ensemble, and as with $MRT_{Wood}$, this loss of ecological information is exacerbated at coarser resolutions. Notably,

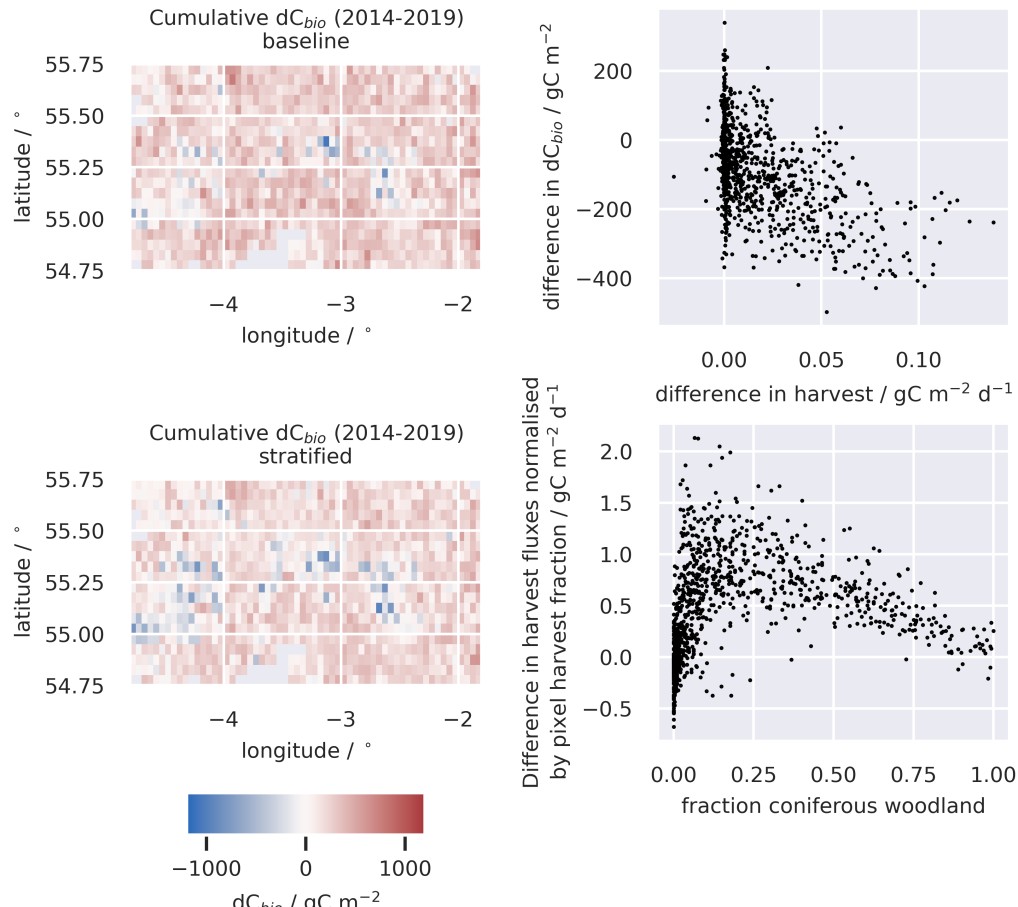

**Figure 6.** A comparison of changes in the C stocks aggregated across the live C pools for the baseline and stratified ensembles for the $0.05°$ domain (left); the relationship between the differences in simulated changes in live C stocks ($dC_{bio}$) and the difference in simulated harvest fluxes for the two approaches (upper right); and the difference in the simulated harvest fluxes, i.e. (stratified - baseline), normalised by the area of the pixel harvested, as a function of the fraction of the pixel covered by coniferous woodland (lower right). Large differences in simulated stock changes were generally associated with corresponding differences in the simulated timber harvest fluxes. The difference between the stratified and baseline harvest fluxes are greater in pixels with a mix of coniferous woodland and other land-cover classes and diminish in more homogeneous pixels. A similar figure for the $0.25°$ domain is provided in the supplementary material (Figure **??**).

there was little variation in leaf lifespan between classes (Figure A7), reflecting the strong seasonal cycle in the assimilated LAI observations for all classes, including coniferous woodland (Figure A2).

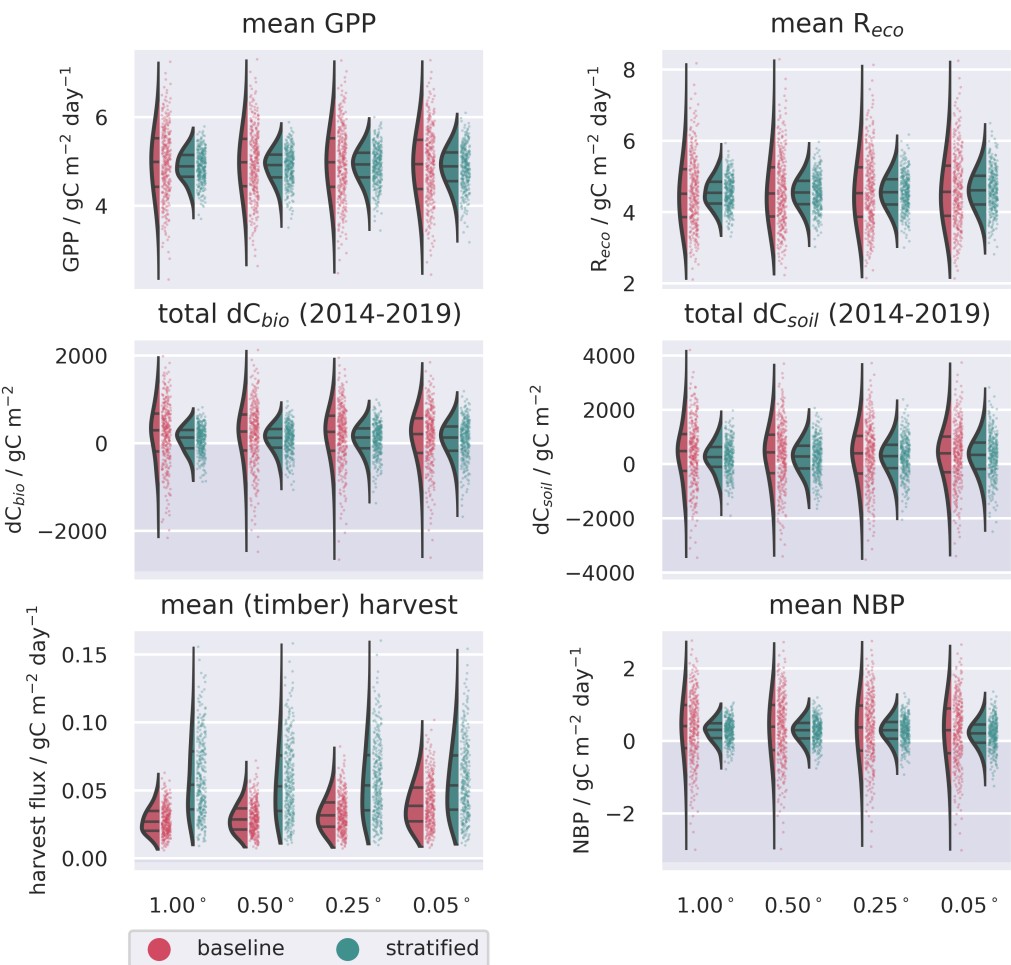

**Figure 7.** Ensemble distributions for the spatially aggregated ecosystem C fluxes and stock changes simulated with the 0.05° resolution domain, contrasting the baseline and stratified analyses. Fluxes represent temporal averages between 2014-2019; stock changes represent cumulative differences over the same period. Uncertainties are assumed to be fully correlated in space, but independent across the different land-cover strata in the stratified ensemble.

Differences in the parameter ensembles calibrated for each stratum lead to contrasting C dynamics in response to future climate change (Figure 9), although the projection uncertainties are large (Figure A10), reflecting the significant role of parameter

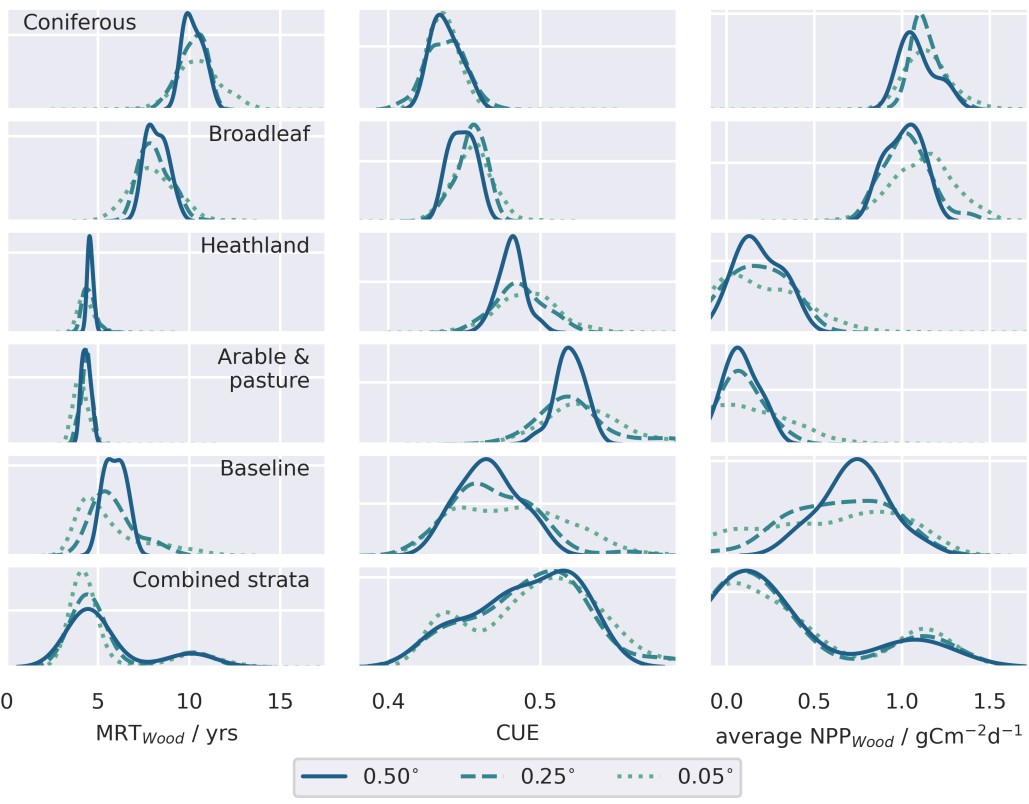

**Figure 8.** Comparison of a subset of retrieved parameters for the individual strata compared to the baseline retrieval for the different resolution domains. Distributions represent the pixel-level median parameter estimates weighted by the pixel-fraction estimated associated with each stratum. Similar distributions of the retrieved pool residence times for the live organic carbon pools and dead organic carbon pools are provided in figure A7 and figure A8 respectively. Allocation fractions of NPP to the live carbon pools are presented in figure A9.

uncertainty in forecasts (Smallman et al., 2021). The forecast simulations do not include any impacts of future disturbance, so the evolution of $dC_{bio}$ post-2020 was driven only by biogenic processes, which were not strongly scale-dependent in our experiments. Both the stratified and baseline ensembles simulated increasing live C stocks up to 2100, assuming no disturbance 390 (i.e. responding to climate and $CO_2$ only), with the rate of growth tailing off in the latter half of the forecast period. While

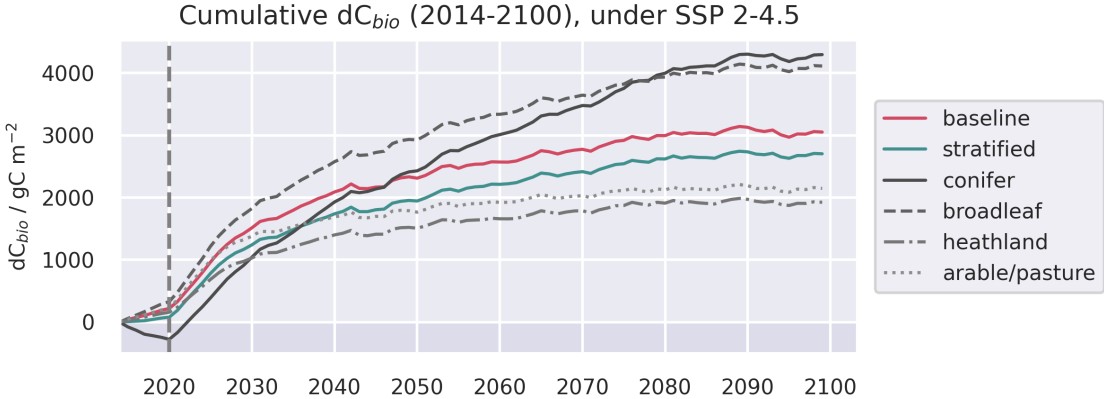

**Figure 9.** Evolution of live C pools aggregated across the baseline and stratified domains, alongside the individual strata, under three future trajectories of climate change. Climate trajectories were extracted from the UK Earth System Model (UKESM; Sellar et al., 2019) contribution to CMIP6 (Eyring et al., 2016). No disturbance was simulated after the end of the calibration period. The presented trajectories were taken from the $0.50°$ domain, and represent the median estimates of the ensemble forecasts.

the trajectories of the baseline and stratified ensembles were similar, the individual strata evolved along divergent paths (Figure 9). The increase, and rate of increase, of live C density ($dC_{bio}$) tracked differences in the retrieved $MRT_{Wood}$ (Figure 8). Accumulation was greatest for the the coniferous and broadleaf forest strata, for which the median $dC_{bio}$ increased throughout the simulation period. In contrast, the heathland class and arable/pasture class accrued C in live biomass more slowly, and had
stopped accumulating C by the end of the forecast scenarios. Of the the two forest strata, the median rate of increase in C density was greatest for coniferous forest (Figure 9).The future trajectories were comparable across the range of spatial resolutions (Figures A11, A12, A13).

## 4   Discussion

### 4.1   Scale-variance of simulated carbon balance in heterogeneous ecosystems, *H1, H2*

Stratification of land-use is a prerequisite to scalable modelling of the C balance due to interaction of uneven distribution of C stocks, and spatially correlated disturbance fluxes. For our study spanning approximately 30,000 km$^2$ of mixed land-use in northern England and southern Scotland, we found contrasting resolution dependence for spatially distributed biogenic fluxes (GPP, $R_{eco}$) and disturbance fluxes, typically restricted to specific land-uses (e.g. timber harvest). In our baseline experiments, where we aggregated the assimilated data to the domain resolution without considering the underlying land-use, GPP and $R_{eco}$
were insensitive to the grid resolution for model domains spanning a resolution range of $1°$ to $0.05°$ (Table 2, Figure A5). While we expected the biogenic fluxes to be less sensitive than exogenous fluxes, the relative invariance in biogenic fluxes with respect to both resolution and method was surprising. Within the version of DALEC used, GPP is estimated for each pixel as a function

of leaf area and meteorology drivers, modulated by the retrieved canopy photosynthetic efficiency parameter. $R_a$ is estimated as a parameterised fixed fraction of GPP, while $R_h$ is proportional to the C stocks in the litter and soil, and exponentially related to temperature. The resolution of the meteorological driving data was 0.5° resolution, so did not resolve differences in the local meteorological conditions across the range of grid resolutions. Additionally, the time series of assimilated LAI did not exhibit strong variations between strata (Figure A2). Together these might explain the relative insensitivity of biogenic fluxes to resolution observed in our experiments. A previous attempt to estimate the impact of meteorological drivers failing to account for local conditions suggested a corresponding error on NEE estimates of ∼10% within 100 km of a met station, compared to an estimated parameter uncertainty of ∼50% (Spadavecchia et al., 2020). Therefore, while fine-scale meteorology may drive fine-scale variability in biogenic C fluxes, we anticipate that the impacts are likely to be secondary to the differences between land cover types within a heterogeneous landscape. However, the influence of fine-scale meteorological variation is an aspect worthy of future research. Our model also ignored changes to litter pools associated with harvest; a more complex treatment incorporating coarse and fine residues might lead to greater sensitivity in $R_h$. Conversely, disturbance fluxes, dominated in this case by timber harvest, showed a clear resolution-dependence, with flux estimates approaching the stratified estimates at the finest grid resolutions (Figure 5), although the baseline estimates did not reach convergence even at 0.05° resolution (Table 2, Figure A5). The emergent resolution-dependence is a consequence of the fact that AGB is not distributed evenly within the landscape, but concentrated in woodlands, areas which unsurprisingly correlate strongly with the distribution of timber harvest. Failing to account for this localisation of disturbance within distinct ecosystems when aggregating to coarse spatial domains therefore results in a significant negative bias in the simulated C losses from the live C pools. In contrast, in the stratified framework, disturbance fluxes were insensitive to resolution, giving a consistent estimate of the carbon balance across the resolution ranges considered.

## 4.2 Impact of heterogeneity on model parameters and ecosystem response to future climate *H3*

We found that by stratifying the landscape prior to MDF, the variability in ecosystem function exhibited between ecosystems, manifest in their retrieved parameters, was retained across the range of spatial resolutions considered (Figure 8). Conversely this ecological information is degraded if data are aggregated without considering *a priori* the underlying distribution of land-use and land-cover, demonstrated by the contraction of the distribution of median parameters across the model domains in the baseline experiments compared to the stratified experiments. Critically, this misrepresentation of ecosystem function was exacerbated at the coarser grid resolutions commonly employed in large scale MDF applications, demonstrated by the contraction of the distribution of median parameters across the model domains. The degradation of ecological information is exemplified by our attempts to constrain mean residence times in the long-lived wood and soil pools, which are critical for understanding the potential carbon sink of terrestrial ecosystems (Luo et al., 2015; Smallman et al., 2021): in the baseline experiment, where data were the relationship between stocks and land-use was ignored, the longer residence times specific to woodland ecosystems ($MRT_{Wood}$; Figure 8) and heathland areas supporting C-rich peat deposits ($MRT_{Soil}$; Figure A8) were not well-represented by the posterior parameter estimates, particularly in the coarser model domains. In this sense, stratification led to the retention of greater ecological fidelity in the model ensemble when aggregating to coarser spatial resolution domains.

The overall ecological fidelity of the model representation will also be limited by the process representation embedded in the model structure, and in the fidelity of the observation data to the relevant characteristics of the actual ecosystem. Prior research has demonstrated that CARDAMOM can retrieve trait differences across biomes (Bloom et al., 2016; Smallman et al., 2021). We demonstrate that CARDAMOM can also retrieve ecosystem-specific traits in mosaic landscapes when the assimilated data is stratified based on prior knowledge of land-use, as far as the observations available for assimilation faithfully convey the ecosystem characteristics. Given the relative importance of parameter uncertainty on future trajectories of the terrestrial C cycle (Smallman et al., 2021), stratification also presents significant opportunities to take advantage of the Bayesian framework embedded within CARDAMOM by taking advantage of the prior information on land-cover and land-use, for example using global trait databases (e.g. Kattge et al., 2020), to inform parameter prior estimates.

In our prognostic experiments, we explored the potential future response of our calibrated ecosystem model under different levels of climate forcing (Figure 9), highlighting divergent future C accumulation, which we suggest is largely related to differences in MRT between strata (Luo et al., 2015; Smallman et al., 2021). Ecosystem responses to climate and disturbance are modulated by their functional traits (e.g. Greenwood et al., 2017). In our experiments, the differences in simulated future C accumulation reflect differences in the calibrated parameters between strata, characterising the differing functional traits of the represented ecosystems. In our study landscape, differences in forecast C accumulation between strata counteracted to give overall trajectories that were similar to the baseline experiments. However, in practical terms, understanding the differential dynamics between ecosystems is likely to be important for the utility of forecasts for land managers and policy makers operating in heterogeneous landscapes (Smallman et al., 2022). Ecosystem-specific calibrations may also be particularly important for understanding the future trajectories of heterogeneous landscapes in regions close to thresholds of abrupt ecological change (Turner et al., 2020).

## 4.3 Limitations of current approach and future work

A key advantage of spatially distributed MDF approaches, such as CARDAMOM, is that model parameters are calibrated locally based on the available observations of the ecosystem. Nevertheless, deficiencies in the assimilated observation streams and/or their uncertainty estimates will propagate to affect the calibrated parameters (Zhao et al., 2020). Excessive seasonality in satellite-derived estimates of LAI have been documented previously in coniferous forests. For example, Heiskanen et al. **?** found that for a conifer forest in southern Finland, the seasonal course of satellite LAI estimates systematically underestimated LAI in the winter months, resulting in exaggerated seasonality compared to local site observations. Likewise, in our stratified experiments, it was notable that the satellite-derived LAI for coniferous woodlands stratum exhibited strong seasonality. As a consequence of the propagation of this systematic error in the assimilated observations, the calibrated leaf lifespans were indistinguishable from deciduous systems (Figure A7). Secondly, repeated estimates of AGB have been demonstrated to significantly reduce uncertainties by helping to constrain the residence times of C within the ecosystem (Smallman et al., 2017). In our experiments we used observations of AGB for two time points, but spaced only a year apart (i.e. 2017,2018; Santoro et al., 2021). The spacing of these estimates is short compared to the residence times of aboveground C in forests and woodlands

(Figure A7. Improved constraints on the residence times and trajectories of the long-lived C pools may potentially be possible with additional, comparable, AGB estimates with greater temporal separation, and reduced uncertainty.

Stratification provides flexibility to improve the process representation for specific ecosystems when applying model-data fusion in heterogeneous landscapes. In the context of the UK, top-down estimates of the terrestrial C balance suggest a pulse of emissions late in the summer, coincident with the main harvest season that is not observed in bottom-up CARDAMOM sim-

480 ulations based on a similarly simple DALEC model structure employed here (White et al., 2019). Stratification by itself does not resolve this discrepancy. In contrast, we find that the temporal patterns of simulated $R_{eco}$ were indistinguishable between the baseline and stratified experiments across all resolutions (Figure A5). However, stratification provides the basic framework within which to add ecosystem sub-models that explicitly model land-use in agricultural settings, for which there are already candidate variants of DALEC. For arable systems, DALEC-Crop simulates the C dynamics associated with the growth, devel-

485 opment and harvest of crops (Sus et al. , 2010); DALEC-Grass models the impact of grazing and mowing in managed pasture (Myrgiotis et al., 2020, 2021). A next step is to integrate these two sub-models into the stratified CARDAMOM framework. While both these DALEC-Crop and DALEC-Grass have been validated at the field scale (Revill et al., 2021; Myrgiotis et al., 2020), validation of the national-scale C balance, where individual pixels contain multiple fields, is challenging. One approach would be to return to the atmospheric inversion estimates (White et al., 2019) and test the extent to which adding this additional

490 process representation resolves the temporal discrepancies during the harvest period.

Finally, in our stratified framework, CARDAMOM retrieves model ensembles that best represent the observations within the calibration period based on a specified set of land-use classes extracted from the LCM2015 land-cover map (Rowland et al., 2017). However, land-use is not static. The terrestrial biosphere is a dynamic environment, with environmental and anthropogenic change driving temporal shifts in land-use and cover that we need to be able to account for. The UK has

495 a relatively stable land-use configuration; however, landscapes with significant land-use change present a challenge, as the current implementation of the MCMC within CARDAMOM assimilates the full time series of calibration data to calibrate a set of time-invariant parameters. This carries the advantage of constraining the model parameters using as many observations as possible. Other frameworks employing sequential approaches to data assimilation provide scope for parameters to shift through time as new data are assimilated, such as four-dimensional ensemble variational data assimilation (e.g. Pinnington et al., 2020)

and particle filters (e.g. Montzka et al., 2011). Adapting CARDAMOM to deal with land-use change is therefore an important target for future development, although such efforts will be reliant on reliable, frequently repeated information on shifting land-use (e.g. Souza et al., 2020). The static parameterisation also presents a limitation when forecasting ecosystem responses to climate change, as we do not account for either adaptive management strategies, or shifts in species composition that may drive functional shifts in the terrestrial C cycle. Consequently, forecasts become increasingly uncertain as the environmental

conditions deviate from the calibration period. Moreover, capturing these complex functional responses to future climate within ecosystem models remains a major challenge (Fisher et al., 2020).

### 4.4 Broader implications for constraining the terrestrial C balance

Landscapes frequently host a mosaic of contrasting land-uses and land-cover types, with heterogeneity exhibited at scales of 10s to 100s of metres. Modelling the land surface carbon balance across large scales requires the aggregation of ecosystems to resolutions that are computationally feasible. Adequately handling this scaling challenge is critical to avoid the introduction of biases into the simulated C balance, particularly when dealing with heterogeneous landscapes. Our results indicate that localised, ecosystem-specific fluxes, such as those related to disturbance, are particularly sensitive to resolution in mixed-use landscapes (Figure 6). Disturbances through logging, clearance for agriculture, or fire are important determinants of the carbon balance in many forest landscapes, globally (Gatti et al., 2021; Harris et al., 2021). Deforestation and degradation are often concentrated at forest edges and coincident with forest fragmentation (Brink et al., 2017; Matricardi et al., 2020). These hot-spots of environmental change are therefore characterised by the juxtaposition of contrasting ecosystems in fragmented landscapes, highlighting the importance of stratification within MDF frameworks constraining the carbon balance in forested regions globally.

The C pools within the comparatively simple C-cycle model structure of DALEC (Bloom and Williams, 2015; Smallman et al., 2017) map well onto the IPCC-recommended pools for reporting greenhouse gas emissions in the Agriculture, Forestry and Other Land-Use (AFOLU) sector (IPCC, 2019). By combining multiple, spatially explicit observation streams with ecological theory embedded in models, MDF approaches such as CARDAMOM ensure conservation of mass balance, ecological "common sense" in the retrieved parameter sets and simulated temporal dynamics, as well as transparent propagation of uncertainties when quantifying ecosystem C fluxes (Smallman et al., 2021). In addition, sub-pixel stratification within CARDAMOM enables sector-specific interrogation of the terrestrial C budget even in mosaic landscapes. For example, in the stratified experiment it is evident that C losses driven by timber harvest over the simulation period result in coniferous woodlands tending to lose C over time. The magnitude and spatial extent of these losses is only readily apparent when we stratified CARDAMOM based on land-use. CARDAMOM, facilitated by land-use stratification, therefore has clear potential to feed into Tier 3 greenhouse gas emissions reporting to the UNFCCC (IPCC, 2019).

## 5 Conclusions

Quantifying the current and future terrestrial C balance is essential to understanding the stability of terrestrial ecosystems facing rapid environmental change, and to support robust national reporting of land-based $CO_2$ emissions. Bayesian MDF frameworks like CARDAMOM integrate the ecological knowledge embedded within models with constraints provided from a range of observation sources and their associated uncertainties, thus providing self-consistent, mass-balanced estimates of systemic C cycling (Luo et al., 2011; Bloom et al., 2016; Smallman et al., 2017). However, when applying MDF across large regions, we are confronted by the challenge of capturing the innate complexity of terrestrial ecosystem with spatial and temporal resolutions that are computationally feasible. This challenge is particularly severe in heterogeneous landscapes with a mosaic of land-uses. Failure to account for sub-pixel ecosystem heterogeneity within MDF inversions leads to bias in flux estimates and degradation of the ecological information embedded within the calibrated model ensembles. We explored the carbon

balance for a region of $\sim$30,000 km$^2$ in the UK using a range of spatial resolutions (0.05° - 1.0°). In our baseline experiment (ignoring sub-pixel heterogeneity), disturbance fluxes in particular exhibited a resolution dependent negative bias that was exacerbated both at coarser grid resolutions and as landscape fragmentation increased. Accounting for fine-scale structure of land-use through through stratification resolved this scale dependence, and yielded higher disturbance fluxes. Stratification also enabled CARDAMOM to retrieve parameter ensembles that preserved the differences in ecological function between different

land-uses.

  Stratification within CARDAMOM therefore provides three key benefits: (i) stratification reduces the scale-dependence of flux estimates, facilitating scaling of CARDAMOM applications across larger spatial domains; (ii) stratification provides transparency for sector-level estimates of the terrestrial carbon balance that could be integrated into Tier 3 national emissions reporting frameworks; and (iii) by separately analysing distinct ecosystems within fragmented landscapes, the loss of ecolog-

550 ical information associated with aggregation to coarse resolutions is limited. Where the observations are accurate and model structure appropriate, this this should improve the ecological fidelity of the calibrated models, enable more robust ecological forecasting, and raises the prospect of mapping spatial variations of ecosystem functional traits based on a diverse range of EO data. Future work will build on this stratification framework to build in more detailed process representation to better account for C fluxes in managed arable (Revill et al., 2021) and pasture landscapes (Myrgiotis et al., 2021). Finally, landscape fragmen-

555 tation and disturbance, whether driven by logging, agriculture or fire, are important determinants of the carbon balance globally. Therefore, while the focus of this study is a temperate landscape within the UK, these results carry broader significance for the application of MDF frameworks to constrain the terrestrial C-balance at regional and national scales.

*Code and data availability.* The driving data, and selected carbon cycle outputs have been archived (Milodowski et al., 2023) at Edinburgh DataShare: https://datashare.ed.ac.uk/handle/10283/4849.

The code to regrid the EO datasets is archived at https://github.com/GCEL/DAREUK_EOregrid.

The code to drive the model and analyse the model output and generate the paper figures is archived at:

https://github.com/GCEL/DAREUK_scale_variance_paper.

The specific version of the CARDAMOM code used for the analysis presented here is archived at

https://github.com/GCEL/DAREUK_CARDAMOM_scale_variance_paper.

Registration to the github repositories is provided on request to either T. L. Smallman or M. Williams.

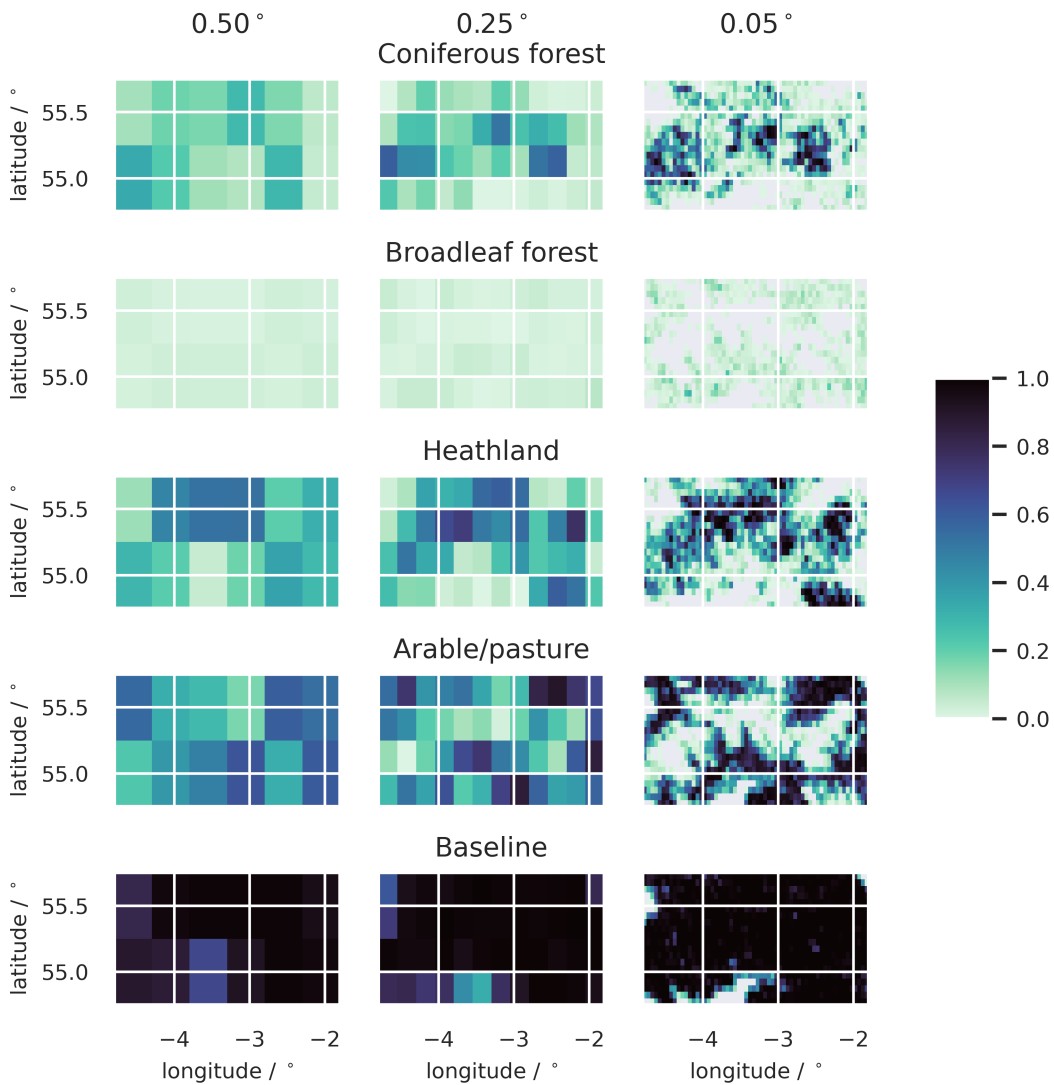

**Figure A1.** Pixel fractions occupied by the different land-cover strata, presented for the $0.50°$, $0.25°$ and $0.05°$ model domains. The $1.00°$ domain is not shown. At $0.05°$ resolution, Recognisable landscape units are picked out with much more clarity.

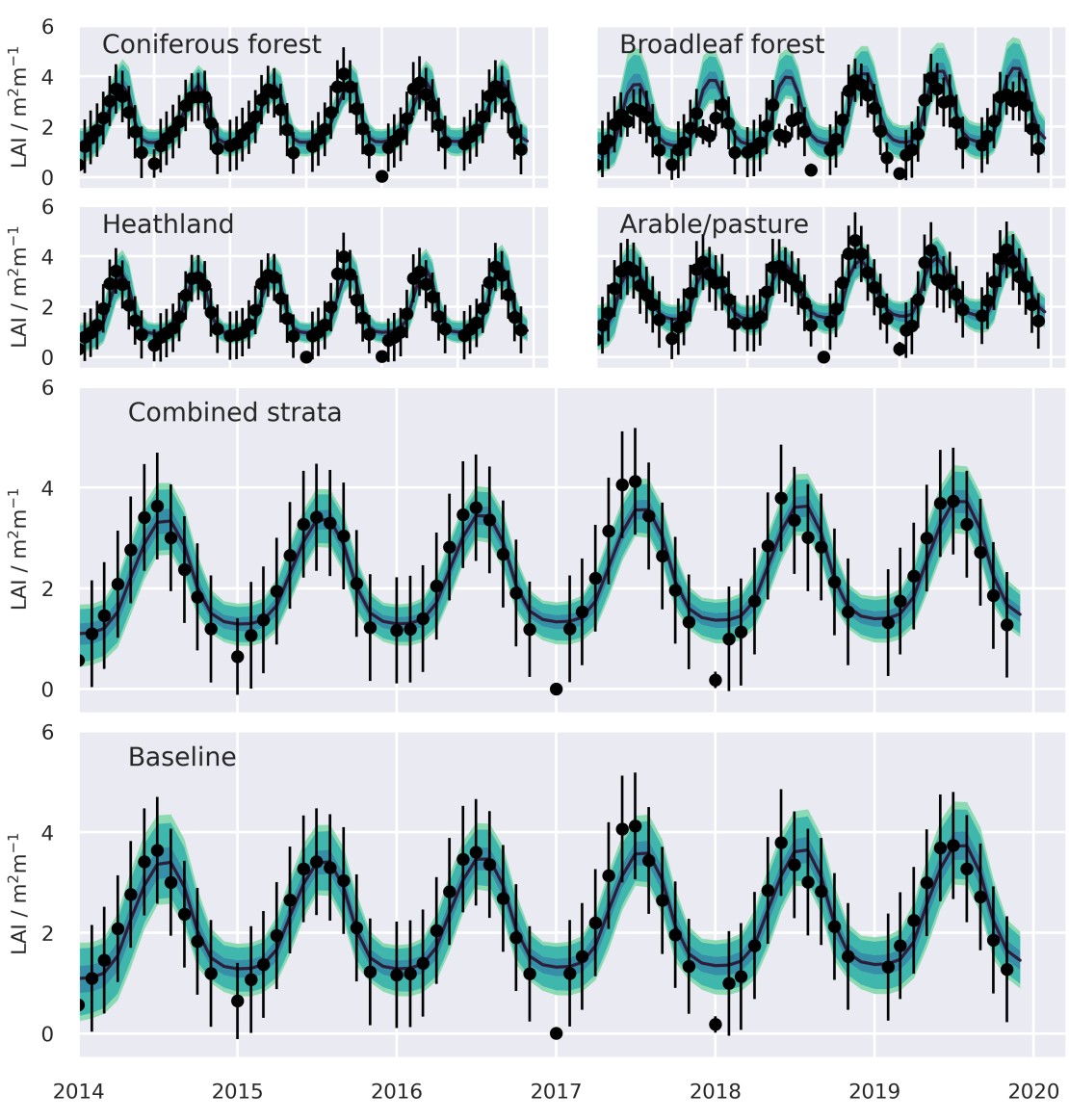

**Figure A2.** Simulated LAI time series and assimilated LAI observations, presented with uncertainty, aggregated from the 0.05° domain. Shaded bands represent the 50%, 90% and 95% confidence intervals, assuming full spatial correlation of uncertainties across the domain. Calibration performance statistics for all resolution domains are summarised in Table 1 and for individual strata in Table A1

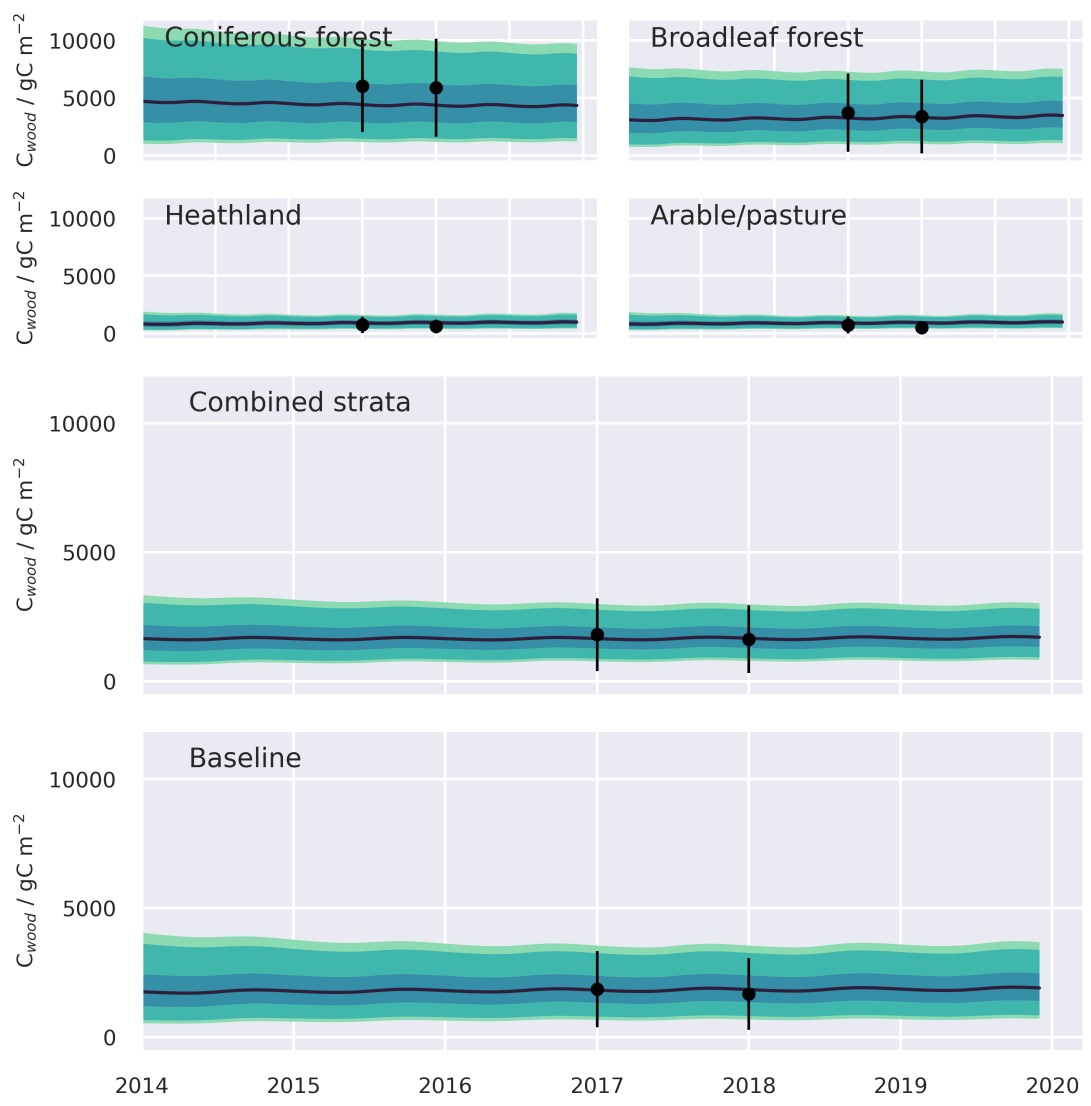

**Figure A3.** Simulated $C_{Wood}$ time series and assimilated observations, presented with uncertainty, aggregated from the $0.05°$ domain. Shaded bands represent the 50%, 90% and 95% confidence intervals, assuming full spatial correlation of uncertainties across the domain. Calibration performance statistics for all resolution domains are summarised in Table 1 and for individual strata in Table A1.

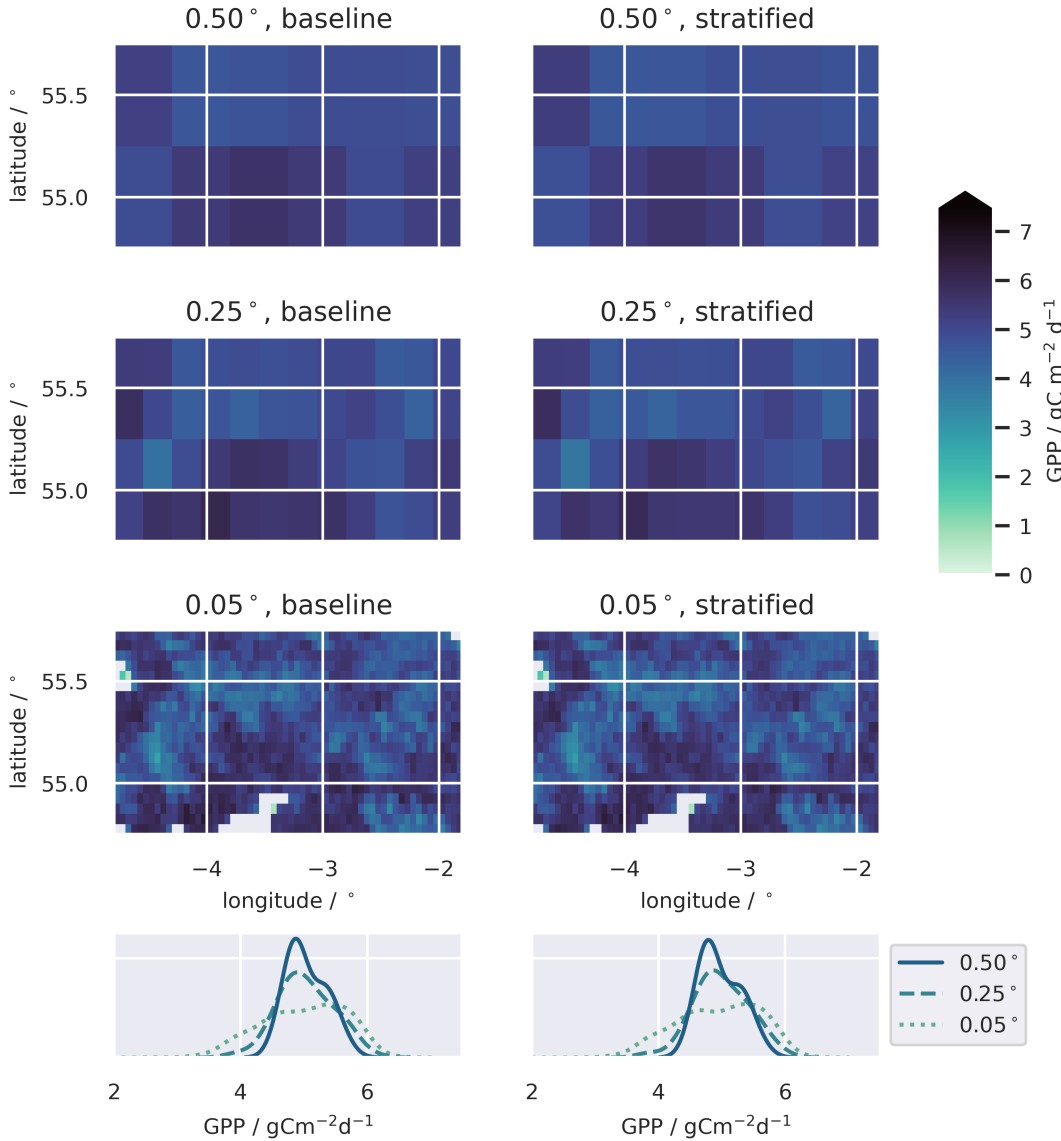

**Figure A4.** Temporally averaged GPP for the baseline and stratified ensembles, for domains of 0.50°, 0.25° and 0.05° resolution. Note the 1.00° domain is not shown. The bottom row shows the distributions of median GPP across the domain for the baseline and stratified ensembles, and demonstrates the diminishing variability in simulated GPP when data are aggregated to spatial domains with coarser resolution.

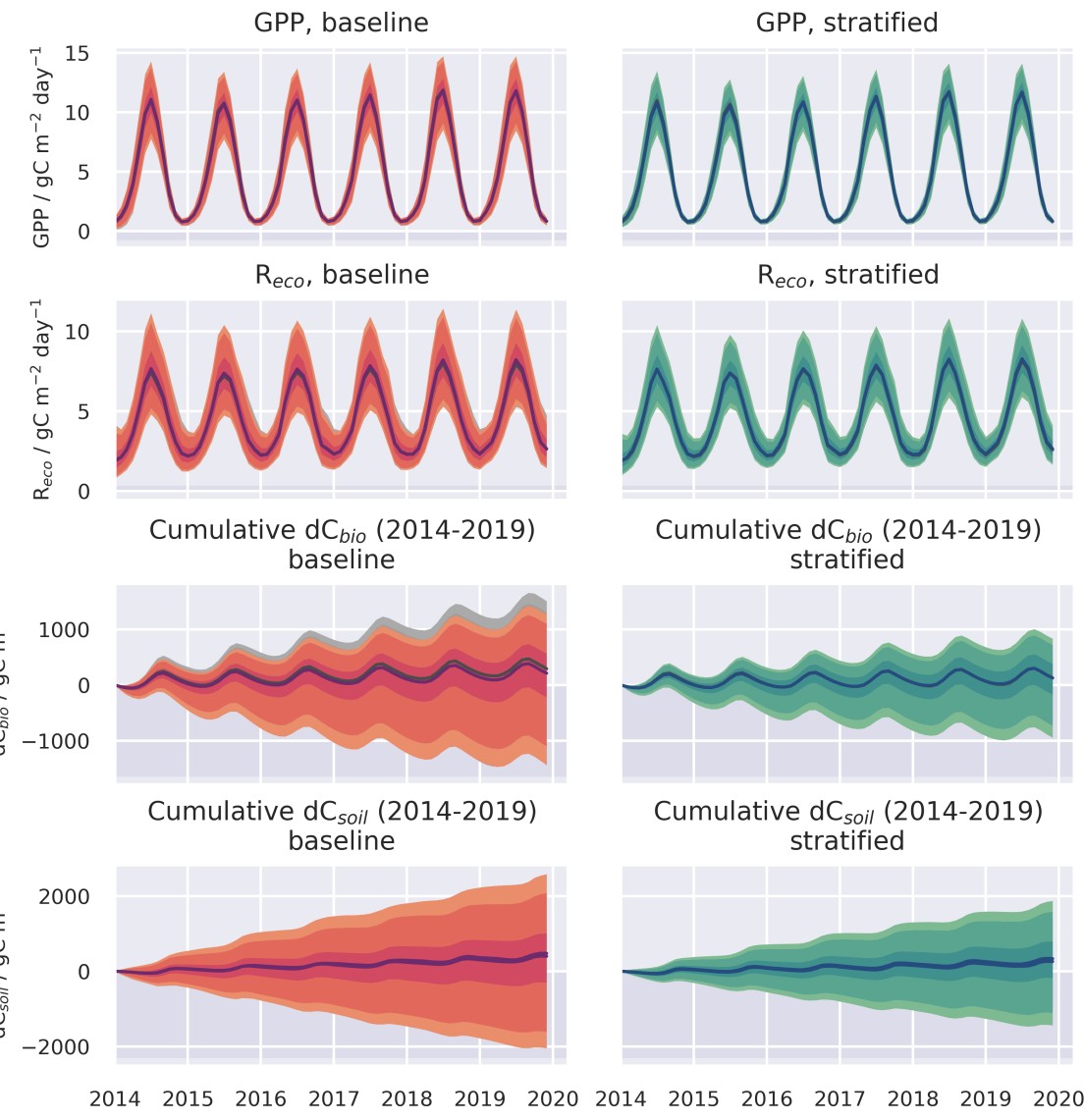

**Figure A5.** Spatially aggregated time series for GPP and ecosystem respiration ($R_{eco} = R_a + R_{het}$), and the cumulative change in carbon stocks in the live ($dC_{bio}$) and soil ($dC_{soil}$) pools, shown for the baseline and stratified ensembles for the $0.05°$ and $1.00°$ resolution domains. The median estimates are plotted with the shaded regions representing the 50%, 90% and 95% confidence intervals. The $1.00°$ simulation results are plotted in grey-scale, but for the most part, the ensemble ranges overlap.

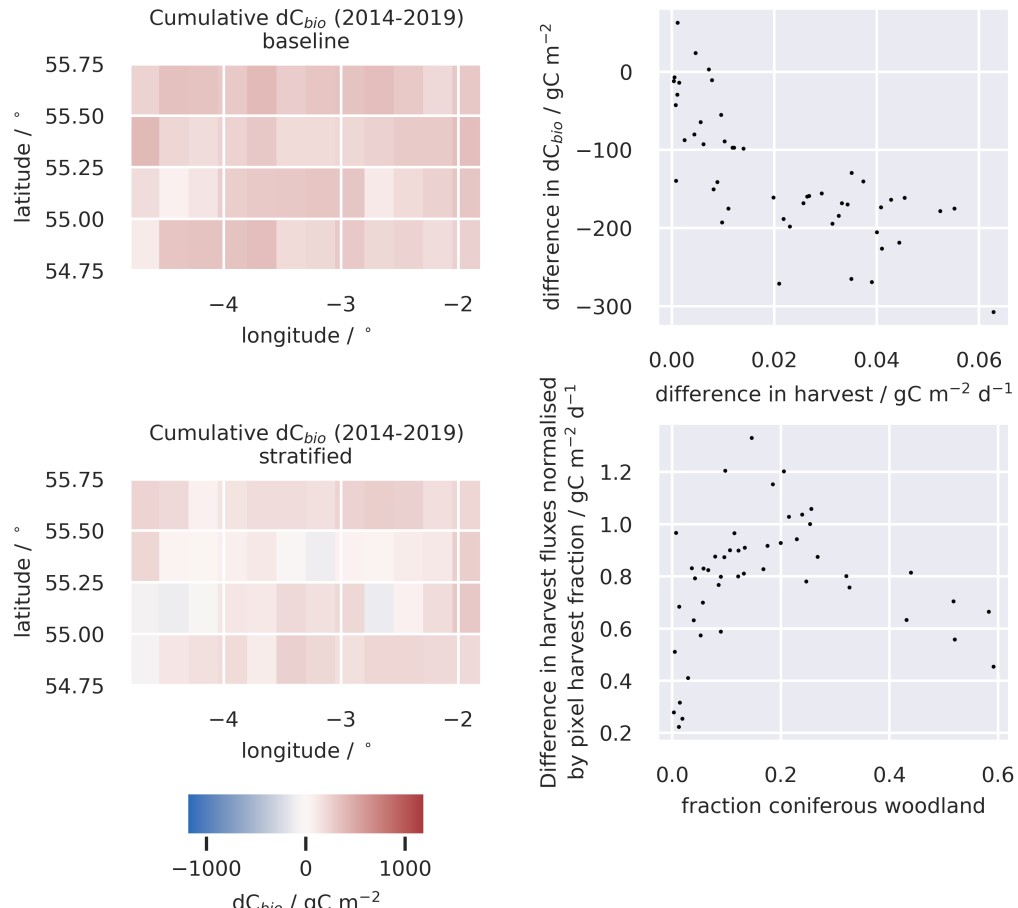

**Figure A6.** A comparison of changes in the C stocks aggregated across the live C pools for the baseline and stratified ensembles for the 0.25° domain (left); the relationship between the differences in simulated changes in live C stocks (dC$_{bio}$) and the difference in simulated harvest fluxes for the two approaches (upper right); and the difference in the simulated harvest fluxes, i.e. (stratified - baseline), normalised by the area of the pixel harvested, as a function of the fraction of the pixel covered by coniferous woodland (lower right). Large differences in simulated stock changes were generally associated with corresponding differences in the simulated timber harvest fluxes. The difference between the stratified and baseline harvest fluxes are greater in pixels with a mix of coniferous woodland and other land-cover classes and diminish in more homogeneous pixels.

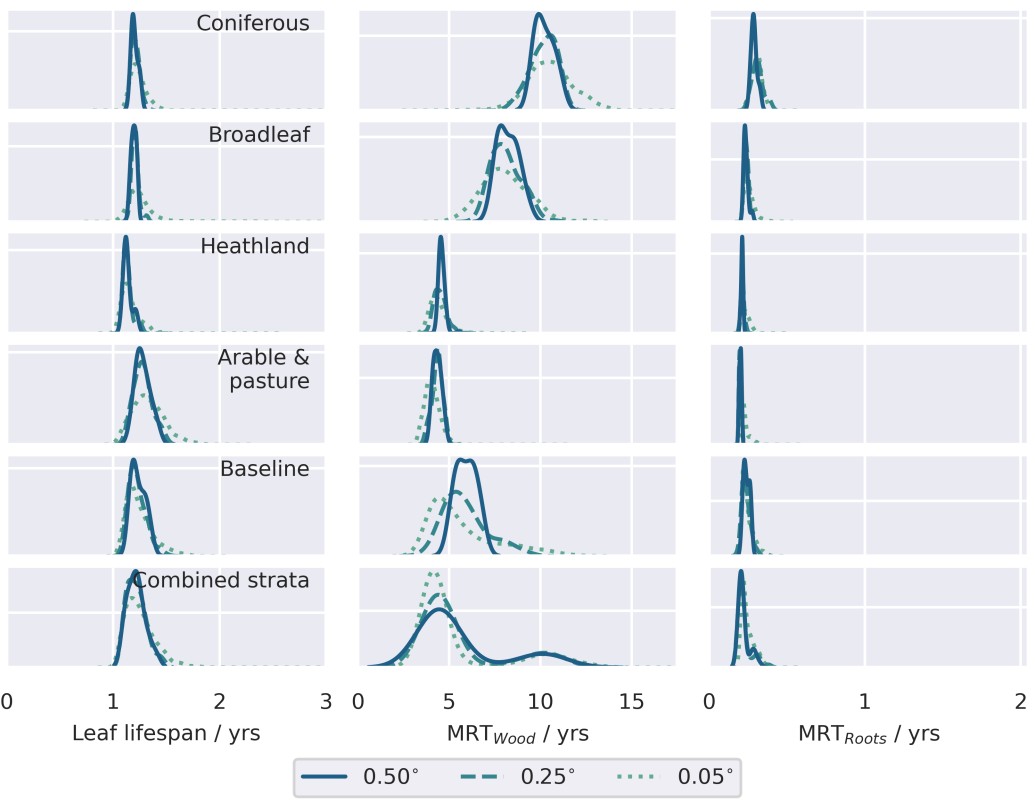

**Figure A7.** Comparison of the retrieved residence times for the live carbon pools, for the individual strata and the baseline retrieval for the different resolution domains. Distributions represent the pixel-level median parameter estimates weighted by the pixel-fraction estimated associated with each stratum.

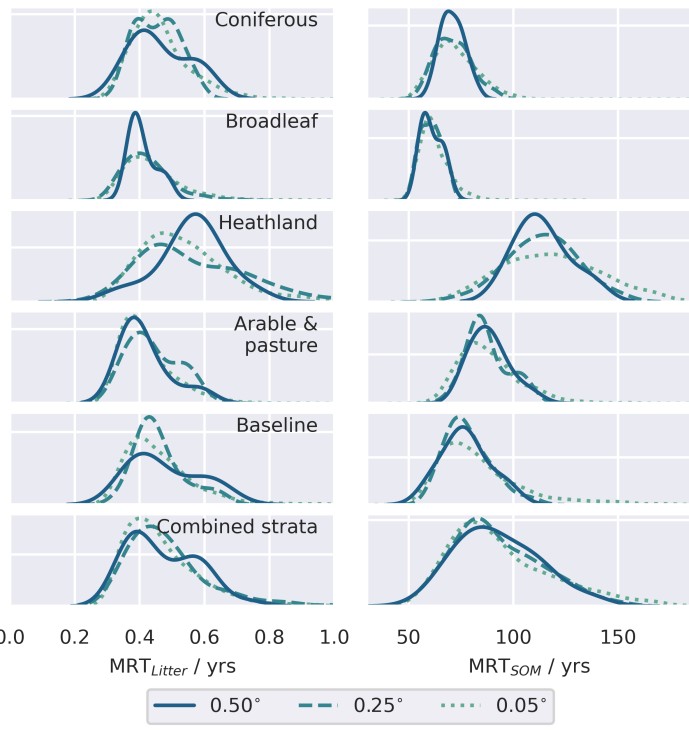

**Figure A8.** Comparison of the retrieved residence times for the dead organic matter carbon pools, for the individual strata and the baseline retrieval for the different resolution domains. Distributions represent the pixel-level median parameter estimates weighted by the pixel-fraction estimated associated with each stratum.

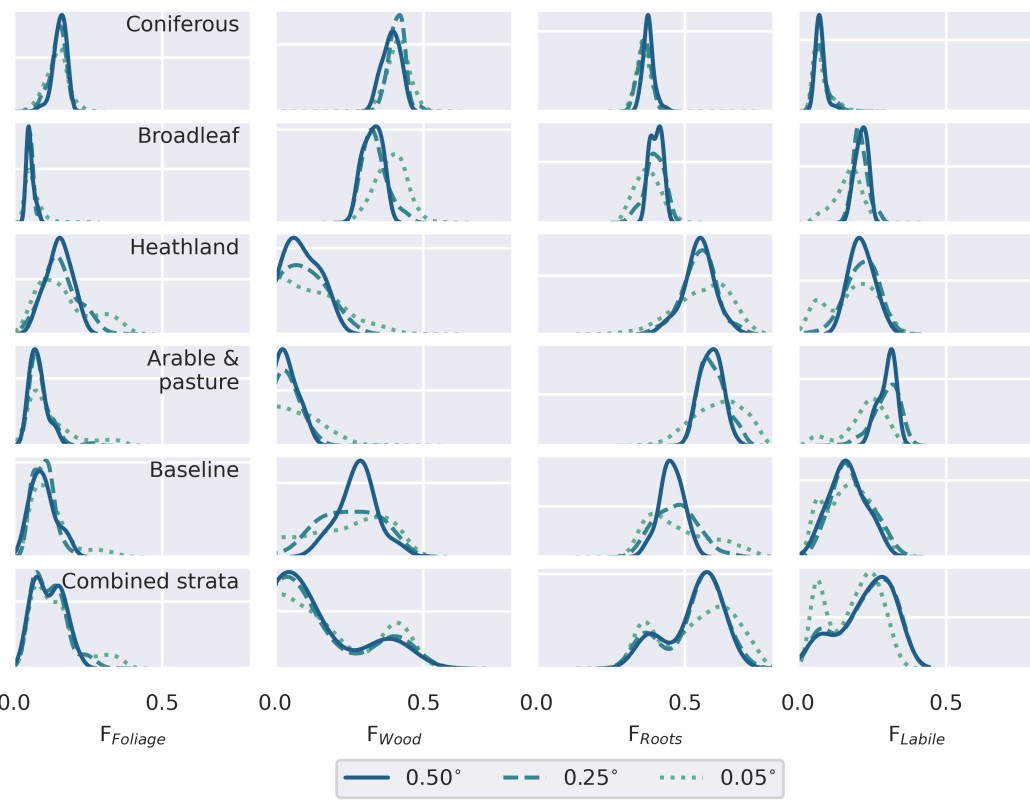

**Figure A9.** Comparison of the retrieved allocation fractions the partitioning of NPP between the live organic carbon pools, for the individual strata and the baseline retrieval for the different resolution domains. Distributions represent the pixel-level median parameter estimates weighted by the pixel-fraction estimated associated with each stratum. Note that labile carbon is subsequently allocated to foliage 7

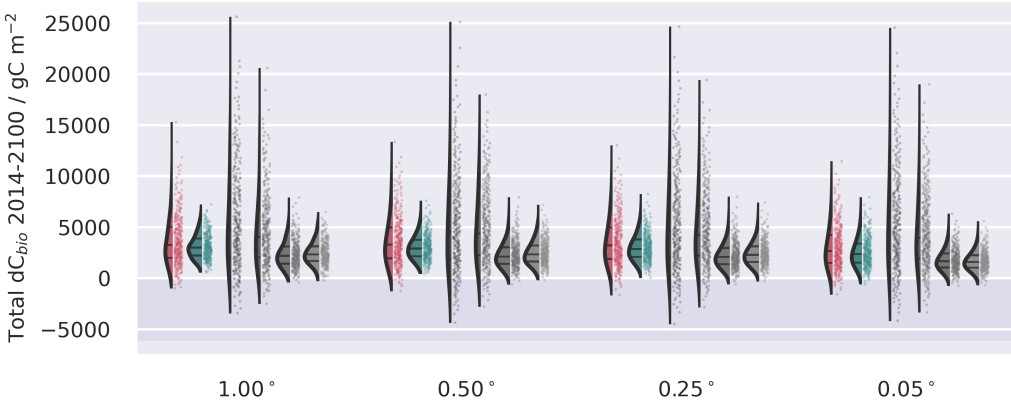

**Figure A10.** Ensemble distributions for the spatially aggregated forecasts of dC$_{bio}$ projected by 2100 under SSP2-4.5W m$^{-2}$ scenario extracted from the UK Earth System Model (UKESM; Sellar et al., 2019) contribution to CMIP6 (Eyring et al., 2016). Stock changes represent cumulative differences from 2016-2100. Uncertainties are assumed to be fully correlated in space, but independent across the different land-cover strata in the stratified ensemble.

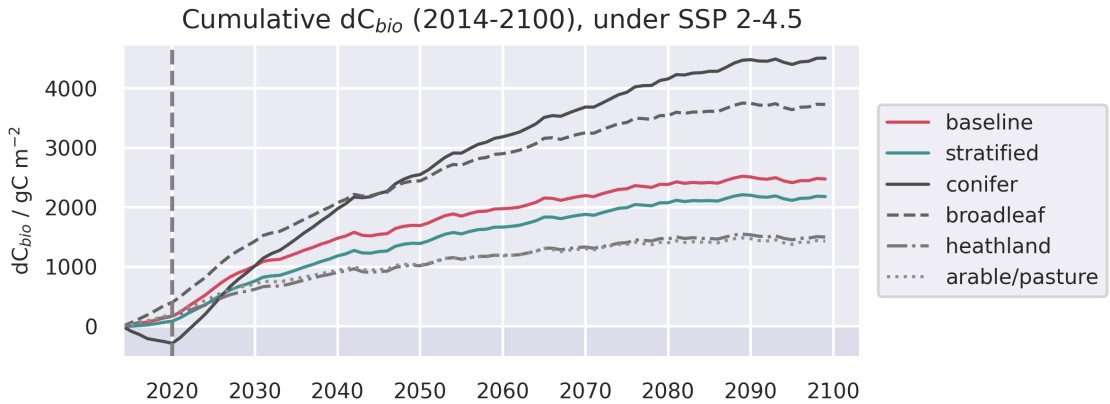

**Figure A11.** Evolution of live C pools aggregated across the baseline and stratified domains, alongside the individual strata, under three future trajectories of climate change. Climate trajectories were extracted from the UK Earth System Model (UKESM; Sellar et al., 2019) contribution to CMIP6 (Eyring et al., 2016). No disturbance was simulated after the end of the calibration period. The presented trajectories were taken from the 0.05° domain, and represent the median estimates of the ensemble forecasts.

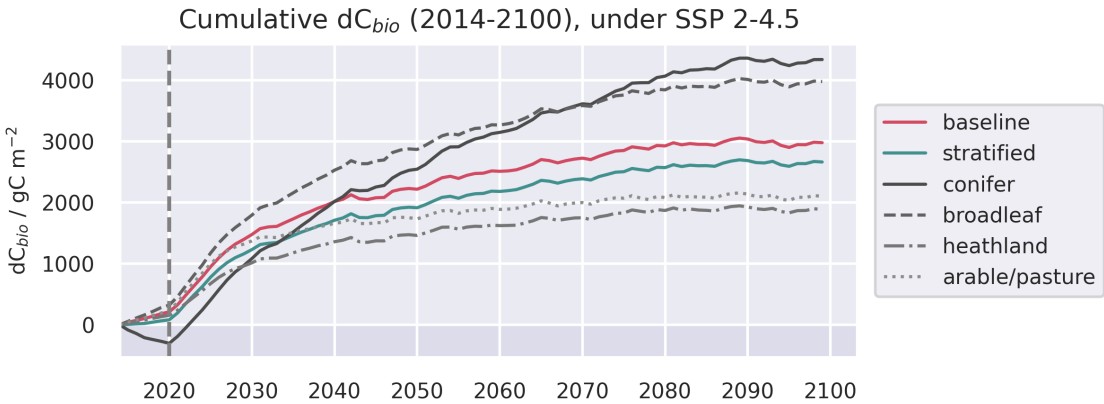

**Figure A12.** Evolution of live C pools aggregated across the baseline and stratified domains, alongside the individual strata, under three future trajectories of climate change. Climate trajectories were extracted from the UK Earth System Model (UKESM; Sellar et al., 2019) contribution to CMIP6 (Eyring et al., 2016). No disturbance was simulated after the end of the calibration period. The presented trajectories were taken from the 0.25° domain, and represent the median estimates of the ensemble forecasts.

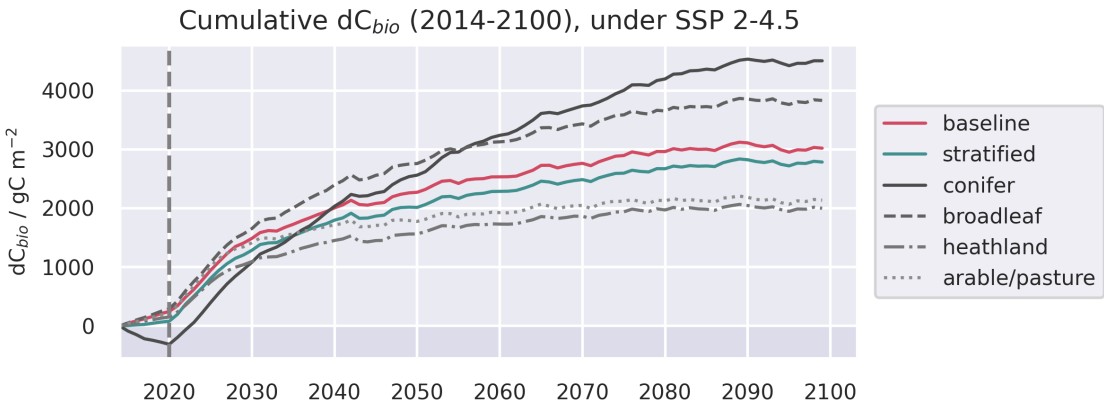

**Figure A13.** Evolution of live C pools aggregated across the baseline and stratified domains, alongside the individual strata, under three future trajectories of climate change. Climate trajectories were extracted from the UK Earth System Model (UKESM; Sellar et al., 2019) contribution to CMIP6 (Eyring et al., 2016). No disturbance was simulated after the end of the calibration period. The presented trajectories were taken from the 1.00° domain, and represent the median estimates of the ensemble forecasts.

**Table A1.** Description of parameters estimated for the DALEC model used in this study (Bloom and Williams, 2015). Each parameter is given a name, unit, description. Note: Gross Primary Productivity = GPP, Autotrophic respiration = $R_a$, Autotrophic maintenance respiration = $R_m$, heterotrophic respiration = Rh. Litter is assumed to be the combined foliage and fine root litter pools. Note that GPP allocation fractions are applied sequentially such that GPP allocation to $C_{wood}$ = GPP - (GPP·$R_a$:GPP) - (GPP·$GPP_{lab}$) - (GPP·$GPP_{root}$).

| Name | units | Description |
|---|---|---|
| $R_a$:GPP | fraction | Fraction of GPP allocated to $R_a$ |
| $GPP_{fol}$ | fraction | Fraction of GPP allocated to foliage |
| $GPP_{lab}$ | fraction | Fraction of GPP allocated to labile |
| $GPP_{root}$ | fraction | Fraction of GPP allocated to fine root |
| Leaf lifespan | years | Maximum natural leaf lifespan |
| Leaf growth day | day of year | Julian day on which max labile turnover to foliage as defined by the phenology model |
| Leaf growth period | days | Standard deviation defining the period over which labile turnover to foliage occurs |
| Leaf fall day | day of year | Julian day on which max foliar turnover to litter as defined by the phenology model |
| Leaf fall period | days | Standard deviation defining the period over which foliar turnover to litter occurs |
| Wood turnover | $day^{-1}$ | Fraction of wood loss per day |
| Fine root turnover | $day^{-1}$ | Fraction of fine root loss per day |
| Litter decomposition | $day^{-1}$ at $0^oC$ | Fraction of fine root loss per day |
| Litter mineralisation | $day^{-1}$ at $0^oC$ | Baseline litter turnover to $R_{het}$ |
| Soil mineralisation | $day^{-1}$ at $0^oC$ | Baseline soil turnover to $R_{het}$ |
| $R_{het}$ coefficient | - | Exponential temperature response coefficient for $R_{het}$ |
| LMA | $g(C)m^{-2}$ | Leaf mass per unit leaf area |
| Ceff | $g(C)m^{-2}day^{-1}$ | Potential photosynthetic activity per unit leaf area |
| Initial labile | $g(C)m^{-2}$ | Size of the labile C pool at time step 1 |
| Initial foliage | $g(C)m^{-2}$ | Size of the foliar C pool at time step 1 |
| Initial fine root | $g(C)m^{-2}$ | Size of the fine root C pool at time step 1 |
| Initial wood | $g(C)m^{-2}$ | Size of the wood C pool at time step 1 |
| Initial litter | $g(C)m^{-2}$ | Size of the litter C pool at time step 1 |
| Initial soil | $g(C)m^{-2}$ | Size of the soil C pool at time step 1 |

**Table A2.** Summary of calibration performance for the individual strata in the stratified CARDAMOM experiment, aggregated across the domains. $\sigma$ represents the uncertainty of the assimilated observation data, thus RMSE / $\sigma$ provides the ratio of the RMSE relative to the uncertainty attached to the observation constraint. The values in parentheses following the RMSE and bias estimates indicate the percentage relative to the mean of the observations across the domain.

| Variable | Version | Metric | 1.00° | 0.50° | 0.25° | 0.05° |
|---|---|---|---|---|---|---|
| $C_{Wood}$ | Coniferous forest | RMSE / gCm$^{-2}$ | 1623 (27.4 %) | 1700 (28.6 %) | 1675 (28.2 %) | 1583 (26.5 %) |
| $C_{Wood}$ | Coniferous forest | RMSE / $\sigma$ | 0.39 | 0.40 | 0.40 | 0.38 |
| $C_{Wood}$ | Coniferous forest | Bias / gCm$^{-2}$ | -1620 (-27.3 %) | -1696 (-28.6 %) | -1670 (-28.1 %) | -1566 (-26.3 %) |
| $C_{Wood}$ | Coniferous forest | Bias / $\sigma$ | -0.39 | -0.40 | -0.40 | -0.37 |
| $C_{Wood}$ | Coniferous forest | Median gCm$^{-2}$ | 4368 | 4279 | 4326 | 4456 |
| $C_{Wood}$ | Broadleaf forest | RMSE / gCm$^{-2}$ | 324 (9.5 %) | 298 (8.7 %) | 318 (9.3 %) | 489 (13.8 %) |
| $C_{Wood}$ | Broadleaf forest | RMSE / $\sigma$ | 0.09 | 0.09 | 0.09 | 0.14 |
| $C_{Wood}$ | Broadleaf forest | Bias / gCm$^{-2}$ | -201 (-5.9 %) | -97 (-2.8 %) | -92 (-2.7 %) | -272 (-7.6 %) |
| $C_{Wood}$ | Broadleaf forest | Bias / $\sigma$ | -0.06 | -0.03 | -0.02 | -0.07 |
| $C_{Wood}$ | Broadleaf forest | Median gCm$^{-2}$ | 3170 | 3273 | 3292 | 3256 |
| $C_{Wood}$ | Heath | RMSE / gCm$^{-2}$ | 303 (44.6 %) | 308 (45.4 %) | 294 (43.4 %) | 276 (41.1 %) |
| $C_{Wood}$ | Heath | RMSE / $\sigma$ | 0.45 | 0.46 | 0.45 | 0.46 |
| $C_{Wood}$ | Heath | Bias / gCm$^{-2}$ | 293 (43.1 %) | 297 (43.7 %) | 279 (41.1 %) | 250 (37.2 %) |
| $C_{Wood}$ | Heath | Bias / $\sigma$ | 0.43 | 0.44 | 0.43 | 0.42 |
| $C_{Wood}$ | Heath | Median gCm$^{-2}$ | 953 | 956 | 934 | 899 |
| $C_{Wood}$ | Arable/pasture | RMSE / gCm$^{-2}$ | 356 (57.2 %) | 360 (57.7 %) | 360 (57.8 %) | 332 (53.3 %) |
| $C_{Wood}$ | Arable/pasture | RMSE / $\sigma$ | 0.57 | 0.58 | 0.59 | 0.56 |
| $C_{Wood}$ | Arable/pasture | Bias / gCm$^{-2}$ | 332 (53.3 %) | 334 (53.6 %) | 332 (53.3 %) | 296 (47.5 %) |
| $C_{Wood}$ | Arable/pasture | Bias / $\sigma$ | 0.53 | 0.54 | 0.55 | 0.51 |
| $C_{Wood}$ | Arable/pasture | Median gCm$^{-2}$ | 937 | 942 | 939 | 907 |
| LAI | Coniferous forest | RMSE / m$^2$m$^{-2}$ | 0.38 (16.2 %) | 0.39 (16.6 %) | 0.39 (16.8 %) | 0.42 (18.0 %) |
| LAI | Coniferous forest | RMSE / $\sigma$ | 0.36 | 0.37 | 0.37 | 0.39 |
| LAI | Coniferous forest | Bias / m$^2$m$^{-2}$ | -0.01 (-0.4 %) | -0.00 (-0.0 %) | -0.01 (-0.3 %) | -0.01 (-0.4 %) |
| LAI | Coniferous forest | Bias / $\sigma$ | -0.01 | -0.00 | -0.01 | -0.01 |
| LAI | Coniferous forest | Median m$^2$m$^{-2}$ | 2.21 | 2.21 | 2.21 | 2.21 |
| LAI | Broadleaf forest | RMSE / m$^2$m$^{-2}$ | 0.47 (17.4 %) | 0.49 (18.1 %) | 0.50 (18.5 %) | 0.54 (22.1 %) |
| LAI | Broadleaf forest | RMSE / $\sigma$ | 0.42 | 0.43 | 0.44 | 0.48 |
| LAI | Broadleaf forest | Bias / m$^2$m$^{-2}$ | -0.03 (-1.0 %) | -0.04 (-1.4 %) | -0.04 (-1.4 %) | -0.05 (-2.2 %) |
| LAI | Broadleaf forest | Bias / $\sigma$ | -0.02 | -0.03 | -0.03 | -0.05 |
| LAI | Broadleaf forest | Median m$^2$m$^{-2}$ | 2.51 | 2.50 | 2.50 | 2.39 |
| LAI | Heath | RMSE / m$^2$m$^{-2}$ | 0.39 (20.5 %) | 0.39 (20.6 %) | 0.41 (21.4 %) | 0.44 (22.8 %) |
| LAI | Heath | RMSE / $\sigma$ | 0.40 | 0.41 | 0.42 | 0.46 |
| LAI | Heath | Bias / m$^2$m$^{-2}$ | -0.02 (-1.3 %) | -0.02 (-0.8 %) | -0.02 (-1.0 %) | -0.02 (-1.2 %) |
| LAI | Heath | Bias / $\sigma$ | -0.02 | -0.02 | -0.02 | -0.02 |
| LAI | Heath | Median m$^2$m$^{-2}$ | 1.77 | 1.78 | 1.77 | 1.77 |
| LAI | Arable/pasture | RMSE / m$^2$m$^{-2}$ | 0.42 (16.2 %) | 0.44 (17.1 %) | 0.46 (17.8 %) | 0.51 (19.7 %) |
| LAI | Arable/pasture | RMSE / $\sigma$ | 0.38 | 0.40 | 0.41 | 0.46 |
| LAI | Arable/pasture | Bias / m$^2$m$^{-2}$ | -0.04 (-1.4 %) | -0.04 (-1.6 %) | -0.04 (-1.6 %) | -0.04 (-1.7 %) |
| LAI | Arable/pasture | Bias / $\sigma$ | -0.03 | -0.04 | -0.04 | -0.04 |
| LAI | Arable/pasture | Median m$^2$m$^{-2}$ | 2.43 | 2.43 | 2.43 | 2.44 |

*Author contributions.*  DTM, TLS and MW created the experimental design. DTM ran CARDAMOM and analysed the outputs. DTM led the writing with inputs from TLS and MW.

*Competing interests.*  The authors have no competing interests.

*Acknowledgements.*  For the purpose of open access, the author has applied a Creative Commons Attribution (CC BY) licence to any Author Accepted Manuscript version arising from this submission. D.T. Milodowski and M. Williams were funded primarily by the NERC DARE-UK project (NE/S003819/1). T. L Smallman and M. Williams were additionally supported by the UK's National Centre for Earth Observation. M Williams also received funding from the Royal Society. This work has made use of the resources provided by the Edinburgh Compute and Data Facility (ECDF) (http://www.ecdf.ed.ac.uk/). LAI information was generated by the Global Land Service of Copernicus, the Earth Observation programme of the European Commission. Aboveground biomass maps for 2017 and 2018 were provided by the European Space Agency through their Climate Change Initiative (CCI) Biomass project. Soilgrids2 soil organic carbon maps were made available by the International Soil Reference and Information Centre (ISRIC). The University of Edinburgh CARDAMOM framework is available on Github (https://github.com/GCEL/CARDAMOM) with permissions provided on request to the corresponding author. The Jet Propulsion Laboratory CARDAMOM framework is also available on Github (github.com/CARDAMOM-framework/CARDAMOM_2.1.6c - contact abloom@jpl.nasa.gov for access).

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
