# Peer review of "Scale-variance in the carbon dynamics of fragmented, mixed-use landscapes estimated using Model-Data Fusion"

_Biogeosciences, 2022_

## Author Response (AR1)

**Reply to RC1**

General comments:

This paper demonstrates the impact of using both finer-scale and categorically refined representations of land surface heterogeneity on modeled carbon stocks and fluxes. The authors present a case study over a region with four dominant land use types to demonstrate that modeling the ecosystem response of each land use type separately, and aggregating the results, does not always yield the same result as modeling the aggregate ecosystem response of the region. The authors document differences in simulated carbon stocks and fluxes, and derived parameters characterizing the ecosystem function, among the different approaches and resolutions tested.

The hypotheses are interesting and well-explored by the experiments chosen, and the results are important. By documenting the sensitivity of the data assimilation framework to the spatial scale and categorization of the data inputs, this work highlights ecological assumptions embedded in standard usage of this and similar models that may undermine their ability to investigate questions of ecological function and future response. Raising this issue is a useful contribution.

> *We thank the reviewer for their detailed review and positive assessment of value of our study.*

However, some of the framing and concluding statements in this paper assert improvements in ecological fidelity or simulated carbon fluxes due to stratification without validating this with any outside data. It would be great if possible to include some validation, such as comparison with outside data to validate derived parameters (e.g., residence times) or carbon/water flux data from flux towers. If this isn't possible, I recommend adopting new language in your framing and conclusions to focus on the sensitivity you demonstrate and make less claim to ecological fidelity or improved representations of carbon fluxes. Section 4.3 is informative and some of the context outlined there could be brought out in the framing and goals at the outset.

> *Validation is challenging at the scale of our model domain; even at the finest spatial resolutions employed here (~5km), there are significant scale differences compared to the scales of site-level observations, such as eddy covariance towers.*
>
> *The key scientific goals for these experiments were to explore the sensitivity of calibrated model parameters and the simulated C fluxes to spatial resolution, to assess directly the scale-dependence of the model-data fusion framework, and to assess the extent to which stratification resolved the emergent scale-dependence while maintaining the spatial resolution of the domain.*
>
> *If spatial resolution systematically biases the model, with weak-to-absent convergence at fine spatial resolutions, then there is an unconstrained, systematic error, and we argue that this indicates a reduction in the ecological fidelity. A qualitative view on whether ecological fidelity has been improved is to assess the retrieved turnover rates between the baseline and stratified cases. We know that forest systems turn over structural carbon at much lower rates than grassland and arable systems. This is reflected in the stratified analysis, but poorly represented in the baseline case. Therefore, we maintain our view that the stratification approach should lead to improved ecological fidelity compared to the*

*baseline, as stratification greatly reduced the scale-dependence in our analysis. That is not to say that there has been no information loss in calibrating the models at a given scale. Other factors affecting the ecological fidelity of the model are model structure, and the quality of the input data. We do not test these aspects here, as this would complicate the interpretation of scale-dependence.*

*While not addressed in the analysis presented here, stratification provides a potential starting point from which to improve the ecological fidelity of both the model structure (i.e. through the inclusion of ecosystem- and management-specific models that better represent ecological processes), model parameters (i.e. using appropriate trait-based constraints, such as leaf lifespan, LMA etc for specific ecosystems), and the observations (i.e. identifying systematic issues and biases, such as the apparently "deciduous conifers", which reflect the seasonal LAI signal in the observations).*

*We accept the reviewer's suggestion that the lack of independent validation limits our ability to claim our models are accurately simulating the terrestrial ecosystem. Throughout the revised manuscript, we have therefore provided a more qualified discussion and conclusions, ensuring we emphasise that model fidelity also relies on additional aspects (i.e. model structure, observation accuracy). However, we do point to a wide range of existing literature using the CARDAMOM approach at both site level and at scale which have been validated using independent observations.*

**Specific comments** are below:

**Comment 1: Section 2.3-2.4:** It would be helpful to see a paragraph at the end of the methods discussing the various spatial scales at play, and how these are integrated into the model pixel in each case. When working with a 0.05 degree (~5km) model resolution but imposing constraints on biomass at 100m, on a soil type at 250m, an LAI at 300m, and a timber harvest at 30m, how are these aggregated across the pixel?

*The stratified approach is as follows:*

- *We aggregate the full classification scheme for the landcover map to match our simplified strata scheme.*
- *For each data product, we regrid the aggregated landcover map to match the product domain, based on the modal class.*
- *For each stratum, for each pixel in our model domain, we then extract all the pixels from the data product that correspond to the stratum in our aggregated (and regridded) land-cover map, and calculate either the mean (AGB, Soil C, LAI) or fraction (tree cover loss, burned area) from this subset.*

*We rewrite these sections to make the methodology clearer:*

*"In the baseline experiments, the observation streams were aggregated to the domain resolutions, and these "community-average" environmental signals were assimilated into a single ensemble. In the stratified experiments, the individual observation streams were stratified at their native resolutions based on the dominant category*

*from the high-resolution land cover map, and then these strata-specific subsets of observations were aggregated to the resolution of the model domains before assimilation."*

**Comment 2:** As a follow on from comment 1, the description of study area emphasizes gradients in temperature and precipitation over topographic features within the 3x3 grid (Lines 128-131), which justifies testing surface resolutions down to 0.05 degree, but then the model runs use 0.5x0.5 forcing data in each case. How do the authors expect this to relate to the amount of scale-dependent variation seen across the model runs?

*The reviewer is correct to identify that the resolution of the meteorological forcing data is 0.5x0.5 deg, so our analysis of scale variance does not account for the changing representation of meteorological conditions across the model domain at differing spatial resolutions. This simplification may underlie the lack of scale-sensitivity exhibited by our simulated biogenic fluxes. A previous study that assessed the impact of meteorological driver error on simulated NEE at a well-studied forest site* (Spadavecchia et al., 2011) *simulated errors in meteorology by interpolating between different combinations of met stations, and found that if the nearest met stations were located <25km away, the contribution to uncertainty on NEE was ~9%, if <100 km away the contribution was 11% and if >100km, then the contribution increased to 17%. In contrast, in their experiments Spadavecchia et al. (2011) found parameter uncertainty accounted for ~50% of the uncertainty in NEE. Therefore, while fine-scale meteorology will likely lead to fine-scale variations in biogenic C fluxes, we anticipate that the impacts are likely to be secondary to the differences between land cover types within a heterogeneous landscape. We will add this discussion to the revised manuscript:*

*"While we expected the biogenic fluxes to be less sensitive than exogenous fluxes, the relative invariance in biogenic fluxes with respect to both resolution and method was surprising. Within the version of DALEC used, GPP is estimated for each pixel as a function of leaf area and meteorology drivers, modulated by the retrieved canopy photosynthetic efficiency parameter. $R_a$ is estimated as a parameterised fixed fraction of GPP, while $R_h$ is proportional to the C stocks in the litter and soil, and exponentially related to temperature. The resolution of the meteorological driving data was 0.5$\circ$ resolution, so did not resolve differences in the local meteorological conditions across the range of grid resolutions. Additionally, the time series of assimilated LAI did not exhibit strong variations between strata (Figure \ref{fig:lai}). Together these might explain the relative insensitivity of biogenic fluxes to resolution observed in our experiments. A previous attempt to estimate the impact of meteorological drivers failing to account for local conditions suggested a corresponding error on NEE estimates of $\sim$10\% within 100 km of a met station, compared to an estimated parameter uncertainty of $\sim$50\% (Spadavecchia et al., 2011). Therefore, while fine-scale meteorology may drive fine-scale variability in biogenic C fluxes, we anticipate that the impacts are likely to be secondary to the differences between land cover types within a heterogeneous landscape. However, the influence of fine-scale meteorological variation is an aspect worthy of future research. Our model also ignored changes to litter pools associated with harvest; a more complex treatment incorporating coarse and fine residues might lead to greater sensitivity in R$\_h$."*

**Comment 3: Section 3.1** The calibration metric, RMSE/sigma, could be further explained. It sounds as though smaller values are desirable here, but if this is a comparison to inherent observational uncertainty, I don't immediately see why <1 is a good thing. Please make this a bit clearer.

*The calibration metric (RMSE/sigma) is intended to highlight the scale of the model-data mismatch relative to the uncertainty in the observations. Our reason for doing so is that the observations themselves often carry large uncertainties, and we feel this metric helps to capture that context when interpreting the quality of the model fit. In terms of whether values <1 are "good", we would argue that values >1 would certainly be undesirable, as this would indicate situations where the model was not able to fit the observations to within their associated uncertainty. We note that where different data streams inform different interpretations of ecosystem functioning (i.e. are the data consistent with each other) there is potential for data streams to provide inconsistent and/or incompatible information (e.g. due to data biases, incorrect specification of uncertainty etc.). This could lead to larger RMSE / sigma ratios as CARDAMOM attempts to balance inconsistent information. Similarly, larger model-data mismatch could also indicate structural error in the ecological representations found in our models. Values <1 may indicate improved constraints based on the assimilation of the complementary data streams (i.e. constraint potentially increases the number and type of observations increase) and the ecological knowledge embedded in the model and EDCs.*

*We add the following into the manuscript for clarification:*

*"Values $>$1 would indicate situations where the model was not able to fit the observations to within their associated uncertainty. Different data streams may provide inconsistent and/or incompatible information, for example due to data biases or incorrect specification of uncertainty citep{zhao2020}. This could lead to larger RMSE/$\sigma$ ratios as CARDAMOM attempts to balance inconsistent information. Larger model-data mismatch could also indicate model structural error. Values $<$1 may indicate improved constraints based on the combination of assimilating complementary data streams and the ecological knowledge embedded in the model and EDCs."*

**Comment 4: Table 1, Table A2:** Additionally, the values of the calibration metric do not proceed monotonically with the shift in resolution. It would be helpful if the authors could explain (or speculate) why this is, especially in the context of the stated goal of improving ecosystem representation by going to smaller scales. A response to this could connect to a response to comments 1 & 2— how does the scale of the input data impact how well things are lining up in the model (applying the right processes to the right initial conditions) at different resolutions?

*The calibration metrics in the table represent the average of the individual pixel level calibration metrics, weighted by the fraction of the pixel covered by the relevant stratum.*

*For the assimilated LAI data, the RMSE increases at finer resolutions consistently at finer resolutions. This result is evident for all land cover strata. The LAI signal carries information relating to phenology, but also to other factors that influence the turnover of leaves, such as disturbances and management processes. The treatment of the canopy dynamics is relatively simple; the same model structure is applied for all strata, and we do not consider ecosystem-specific and/or management-specific variants that would be*

*better adapted to simulating e.g. arable systems. Therefore the canopy dynamics encoded in the model have limited ability to fit related high temporal frequency variations in the canopy dynamics related to disturbance and management that may emerge at finer spatial resolutions. When aggregating over larger areas, complexities in the dynamics of canopy turnover are averaged out. Averaging over larger areas also reduces the noise in the LAI signal. We expect that these factors together underly the increase in RMSE for LAI at finer resolutions. For the assimilated C$_{Wood}$ stocks, there are only two data points. While there is a general trend towards higher RMSEs for the finer resolution domains, this is not consistent for all land cover classes (e.g. conifer woodland). The reason for this is not clear. We note that for each pixel, there are only two data points, spaced a year apart, compared to (potentially) monthly assimilated data for LAI. There are therefore fewer dynamic constraints on wood in the calibration.*

*In terms of the scale differences between data sources, in the ideal circumstance, the input data would match the resolution of the land cover data on which we base the stratification, assuming that the signal-noise ratio is not adversely impacted. For example, the 300m resolution LAI data will not resolve edges as precisely as the 100m AGB data, which itself will not precisely resolve the edges present in the 25m resolution input data. This scale mismatch presents a potentially source of observation error that will be more significant in smaller landscape fragments.*

*One apparent advantage for stratification is that we can sample the data at fine resolutions, but aggregate the sampled data over larger areas to increase the signal:noise ratio without averaging across functionally distinct ecosystems. In our experiments this did not lead to additional biases associated with running analysis on coarser grids, unlike the baseline case. In practise, the selection of resolution would need to trade off the beneficial effects of aggregation on enhancing signal:noise and computational expense, with retaining sufficient spatial resolution to address the research questions that are of interest. Alternative approaches to aggregating the strata into potentially irregular clusters of landscape "patches", rather than our grid-based approach, could be one way to limit the effect of very small area coverage on the analysis, but this would have been challenging to add into our experimental setup while still providing comparison against the baseline experiments across grid resolutions. We note that this "patch" based approach has been used with CARDAMOM for field-scale analysis of pasture (e.g. Myrgiotis et al., 2022) and arable systems (e.g. Revill et al., 2021).*

*We have added the following into the results section (3.1) where the calibration results are presented:*

*"For both LAI and C$_{Wood}$, the RMSE tended to increase at finer spatial resolutions in both the baseline experiments and the aggregated stratified experiment (Table \ref{table:calibration}), although the resolution-dependent trend was not consistent between individual strata, Table \ref{table:strata_cal}).  An increase in RMSE at finer spatial resolutions can be rationalised by the smoothing effect of aggregating the remote sensing products over larger spatial scales. This not only removes the impact of high frequency random noise in the assimilated signal, but also removes variability generated by local processes (e.g. management) that are not accounted for in the relatively simple treatment of canopy dynamics encoded in the model."*

**Comment 5: Line 275 and Figure A2:** Please explain why the coniferous woodland has a strong seasonal cycle of LAI which reaches zero in the winter. The black dots in Figure A2 top left panel suggest that this oscillation is present in the earth observation data, but Scotland is not known for its deciduous conifers. Does snow blanketing the tree canopies, masking out the greenness or making the canopies indistinguishable from the ground, cause this seasonality in the Copernicus LAI product, or is this considered ecologically realistic for your region? If it is unrealistic, does this matter to the resulting biomass trends— for instance, did the authors test a different (presumably more realistic) oscillation bottoming out at ~3?

*The reviewer raises the issue of the incongruity of strongly seasonal LAI over the conifer forests in our domain. Excessive seasonality has been shown to be a significant issue for satellite-based estimates of LAI in conifer forests (e.g. for boreal forests, Heiskanen et al., (2012) highlight the issue for MODIS). We do not specify the phenology of the target ecosystem, but allow it to emerge through the calibration, and thus the seasonal observational LAI gives rise to seasonal simulated LAI dynamics.*

*We did consider the option of applying alternative strategies for assimilating LAI for conifer systems, but doing so would require treating one stratum differently and therefore potentially confound the scale-variance impacts that are the target of the paper. Of course, stratum-specific adaptions of the model-data fusion framework, including modifications relating to the assimilated data, relies on prior stratification based on the ecosystem, and therefore is a potential advantage to our stratified approach. We acknowledge the reviewer's comment and will make it clear in the manuscript that systematic issues associated with the input data will propagate through the analysis. But that addressing data quality challenges in our inputs is beyond the scope of the stated objectives and hypotheses of this paper.*

**Comment 6: Section 3.3:** Regarding the differential response to disturbance flux, it would be helpful if the authors emphasize earlier on that this arises from a mismatch in applying the disturbance to the correct land cover type when using the aggregated pixels. I see the authors do come to this in lines 358-361 but would appreciate it earlier. Section 2.3.5 could be a good place to explain how the authors imposed the disturbance flux in each case, so it is ultra-clear why this difference in how the disturbance is allocated to each land use type arises between the two cases.

*We appreciate the suggestion and will ensure that this is clear in the updated manuscript during the methods description:*

*"To convert area estimates of tree cover loss into changes in C stocks, we use a simple clearance model in which a fraction of the C stored in $C\$\_{wood}\$$, $C\$\_{foliage}\$$ and $C\$\_{labile}\$$ is removed based on the pixel fraction (or stratum-specific subpixel fraction) identified within the GFW dataset as experiencing tree cover loss. In practise, most tree cover loss occurs in the conifer woodlands, and is therefore concentrated in these woodlands in the stratified analysis, compared to the baseline experiments, in which we do not consider the sub-pixel distribution of land cover."*

**Comment 7: Section 3.3:** Line 319: "it is evident that stratification leads to preservation of ecological information across resolutions" and similar statements throughout; suggest to qualify these statements, e.g., "as encoded in the observations available to the model". Getting back to

the conifers acting like deciduous trees— it is important to tread carefully with caveats that the observations themselves come with many assumptions, and may not always represent ecological fidelity. The authors make this caveat in section 4.3 line 395-400, but it would be helpful to keep it at the forefront throughout.

> *We thank the reviewer for this helpful suggestion, and will qualify statements regarding ecological fidelity to reflect that limitations to fidelity will be imposed by both issues with model structure (e.g. missing processes, such as harvest in arable landscapes) and observations (e.g. our "deciduous" conifer woodlands).* For example, *this section of the results now reads:*
>
> *"The stratified data assimilation scheme reveals emergent differences between ecosystems, while traits retrieved for the baseline experiments characterised intermediate values (Figures …). In the baseline experiments, comparing across domain resolutions, it is apparent that aggregation to coarser spatial resolutions reduces the range of retrieved traits. The reduction in the widths of the retrieved parameter distributions highlights the loss of information relating to variations of fundamental aspects of ecosystem function in the baseline experiments as the resolution of ecological gradients is lost. In contrast, it is evident that stratification leads to a reduction in the extent of ecological information loss when aggregating to coarser resolutions, as the widths of the aggregated parameter distributions maintained."*
>
> *Additionally, in the conclusions, the specific reference to ecological fidelity has been modified from*
>
> *"(iii) by separately analysing distinct ecosystems within fragmented landscapes, the ecological fidelity of the calibrated model parameters is enhanced, enabling more robust ecological forecasting and raising the prospect of mapping spatial variations of ecosystem functional traits based on a diverse range of EO data."*
>
> *to:*
>
> *"(iii) by separately analysing distinct ecosystems within fragmented landscapes, the loss of ecological information associated with aggregation to coarse resolutions is limited. Where the observations are accurate and model structure appropriate, this this should improve the ecological fidelity of the calibrated models, enable more robust ecological forecasting, and raises the prospect of mapping spatial variations of ecosystem functional traits based on a diverse range of EO data"*

**Comment 8: Section 4.3 line 395-320:** Great points. It would help satisfy reader curiosity if the authors could delve a bit into these deficiencies (Zhao et al., 2020; Heiskanen et al., 2012) as relevant to the datasets they are using, and discuss how the deficiencies might impact their results. This comment has substantial overlap with comment 5.

> *Zhao et al. (2020) compared LAI estimates from a number of global satellite LAI products against temporal field observations across a suite of contrasting forest sites in China. They found that significant discrepancies between the two, and importantly found that the uncertainty estimates associated with the satellite-derived LAI products were not sufficient to capture the difference between the satellite estimate and field-based estimates.*

*Perhaps most directly relevant to the case of overly-seasonal LAI is the study by Heiskanen et al. (2012), who compared the satellite-based LAI phenology from MODIS against field estimates and found that the satellite estimates the seasonality in the MODIS product was too great, with underestimation of LAI in the winter. We find the same situation for satellite LAI estimates across the conifer woodlands in our region. The reason underlying this issue remains uncertain. The impact of assimilating systematically erroneous data will be to train calibrated parameters that attempt to fit the systematic errors. For our coniferous woodlands, this means calibrating an effectively deciduous canopy, with impacts across the relevant parameters (leaf lifespan, LMA, etc.). Of course, if we know a priori that the land cover is coniferous woodland, we can modify the assimilation scheme, for example assimilating only the summer LAI and facilitating the inversion through priors on canopy traits, such as the expected leaf lifespan. In our current CARDAMOM framework, we do not make any attempt to correct for systematic data issues. Our reason for doing this is it would complicate the comparison of the stratified analysis against the baseline. However, this is very much an avenue we believe will be productive for future research. We will develop this aspect in the revised manuscript.*

*"Excessive seasonality in satellite-derived estimates of LAI have been documented previously in coniferous forests. For example, Heiskanen et al. (2012) found that for a conifer forest in southern Finland, the seasonal course of satellite LAI estimates systematically underestimated LAI in the winter months, resulting in exaggerated seasonality compared to local site observations. Likewise, in our stratified experiments, it was notable that the satellite-derived LAI for coniferous woodlands stratum exhibited strong seasonality. As a consequence of the propagation of this systematic error in the assimilate observations, the calibrated leaf lifespans were indistinguishable from deciduous systems (Figure \ref{fig:residence_times_live})."*

**Comment 9: Line 406-415:** This is a very strong point. It would be great to go further and see the authors chart out a bit what is needed to actually do these improvements in process representation— what is the to-do list? How will improvements be verified?

*We are currently adapting our framework to include versions of the specific arable and pasture models cited in this section. This will improve the process representation of management in these ecosystems.*

*In terms of the "to-do" list, the arable and pasture version of DALEC have already been developed (Sus et al., 2010; Revill et al., 2021), and for pastures, a UK-wide analysis has recently been published (Myrgiotis et al., 2022).*

*To validate the integrated framework, we aim to do a comparison against independent top-down (i.e. atmospheric inversion) estimates of the biogenic flux from atmospheric inversions, following the approach published by White et al. (2019). In this study, the atmospheric inversion of atmospheric $CO_2$ observations from the UK surface tall-tower network indicated that the UK terrestrial biospheric C balance exhibits an increase in the net-flux of C to the atmosphere in the late summer/early autumn, and that this signal was not captured by the version of DALEC used (comparable to our baseline model). We expect that improving arable and pasture representation will help reconcile this major temporal discrepancy between top-down and bottom-up approaches.*

*The revised paragraph is as follows:*

*"Stratification provides flexibility to improve the process representation for specific ecosystems when applying model-data fusion in heterogeneous landscapes. In the context of the UK, top-down estimates of the terrestrial C balance suggest a pulse of emissions late in the summer, coincident with the main harvest season that is not observed in bottom-up CARDAMOM simulations based on a similarly simple DALEC model structure employed here* (White et al., 2019). *Stratification by itself does not resolve this discrepancy. In contrast, we find that the temporal patterns of simulated R$_{eco}$ were indistinguishable between the baseline and stratified experiments across all resolutions (Figure \ref{fig:time_series_medians}). However, stratification provides the basic framework within which to add ecosystem sub-models that explicitly model land-use in agricultural settings, for which there are already candidate variants of DALEC. For arable systems, DALEC-Crop simulates the C dynamics associated with the growth, development and harvest of crops* (Sus et al., 2010); *DALEC-Grass models the impact of grazing and mowing in managed pasture* (Myrgiotis et al., 2022, 2020). *A next step is to integrate these two sub-models into the stratified CARDAMOM framework. While both these DALEC-Crop and DALEC-Grass have been validated at the field scale* (Revill et al., 2021; Myrgiotis et al., 2020), *validation of the national-scale C balance is challenging. One approach would be to return to the atmospheric inversion estimates* (White et al., 2019) *and test the extent to which adding this additional process representation resolves the temporal discrepancies during the harvest period."*

And a few minor **technical comments** below:

**Comment 10: Table 1** last CWood stratified row, shifted numbers

*Thank you for pointing this out. We will fix this in the next version.*

**Comment 11: Figure 6:** What does the color of dots (blue vs green) in the right panels mean? A legend would help

*We have updated the figures.*

**Comment 12: Figure A2-A3:** A legend would help here also.

*We have updated the figures*

**Reply to RC2**

This paper demonstrates the importance of accounting for fine-scale structure in heterogeneous landscapes to ensure ecological fidelity in modeling carbon dynamics. The authors designed two different approaches with model-data fusion to constrain estimates of model parameters and their uncertainty, and compared the difference in simulated carbon dynamics by these approaches with varying spatial scales over a mixed land-use region of the UK. The paper is overall well-written, and the experiment is well-designed. However, I have serval concerns regarding the methods and conclusions. Please see my comments below.

*Thank you for your detailed review and positive comments regarding the manuscript. We address your comments below.*

Major concerns:

The model-data fusion framework (CARDAMOM) assimilated remotely sensed LAI and aboveground biomass (line 158), but soil organic carbon (SOC) extracted from SoilGrid2 was "used to set a prior constraint on the initial SOC stock. Is SOC a constraint in CARDAMOM? To set the initial SOC stock, how did the initial SOC be set for the baseline and stratified cases and for each spatial scale? Did the authors separately set the initial SOC for each sub-pixel type and constrain the sum/average of sub-pixel types to the SOC value derived from SoilGrid2? Please provide more information about this.

> *Soil organic C is one of the C pools represented in DALEC. CARDAMOM provides the option to give prior estimates, with an associated uncertainty, for the initial C stocks in each of the pools. If no constraint is given, a uniform prior is used.*
>
> *The specific method for generating the prior SOC stock for each stratum in the analysis followed the same method applied to extracting data from the other assimilated data streams. The difference is that because there is no date associated with the SoilGrids2 dataset we use these estimates, with their uncertainty, to provide a prior constraint on the initial Soil C stocks.*
>
> *We will ensure that this is communicated more clearly in the revised manuscript:*
>
> *"As there is no date associated with the SoilGrids2 dataset we use these estimates, with their uncertainty, to provide a prior constraint on the initial SOC stocks. This contrasts with our treatment of the LAI and AGB data, which are associated with specific time periods and therefore used as observational constraints on the simulated time series. However, the aggregation of the original SoilGrids2 data layers for the baseline and stratified experiments follows the same procedure."*

Is Cwood in the Results section the aboveground biomass or total woody carbon that includes both aboveground and belowground? Line 190, "Cwood pool is also a reservoir for non-woody structural tissues, for example in areas covered by crop and pasture." Does DALEC also have a woody carbon pool for pasture? Did the authors also infer belowground woody carbon for crop and pasture based on the allometric relationship in Eq. 3?

> *The version of DALEC employed for all strata is the same, with four live pools representing the labile, foliar, fine roots and wood pools. The belowground woody carbon is estimated using the same equation for all strata. We avoided using different approaches for specific stratum as this would have confounded the investigation of scale-dependence in the two approaches, which was the primary target of the experiments. We will make this reasoning clear in the revised manuscript.*
>
> *We note here that while not employed in these experiments, a stratified approach does provide the potential to improve the ecological fidelity of the model structure for specific land-use classes in the future, such as the use of ecosystem-specific models for arable and pasture systems, as discussed in Section 4.3. This is a target for our current work. To improve the clarity of the manuscript, we have amended this section, which now reads:*

*"For simplicity of comparison across the experiments in this study against the baseline experiments (i.e. no stratification), we use only one model structure across all strata, and pre-process the assimilated data streams in the same way. For strata where woody tissues are not part of the dominant vegetation types, for example in areas covered by crop and pasture, the $C_{Wood}$ pool also provides a reservoir for non-woody structural tissue, with the differential allocation patterns and turnover rates reflected in the retrieved parameters. Importantly, different ecosystems could in the future be modelled with distinct, ecosystem-specific models that better capture their functional process dynamics. Relevant ecosystem-specific model variants have previously been integrated within the CARDAMOM framework, for example woodlands* (Smallman et al., 2017)*, pasture* (Myrgiotis et al., 2022) *and arable agriculture* (Revill et al., 2021)*. Given the computational limitations on the resolution of the model domain, stratification would be prerequisite to the inclusion of ecosystem-specific models within regional CARDAMOM applications."*

In table 1, it seems that Cwood and LAI are underestimated (i.e., negative bias) in both baseline and stratified cases across spatial scales. What are the possible reasons for it? Maybe the parameters were not fully constrained?

*The uncertainties associated with Cwood are large. We ran an experiment whereby we artificially reduced the uncertainties by 10%. In this scenario, CARDAMOM retrieved unbiased estimates of Cwood. Thus we conclude that the negative bias between the median retrieved Cwood in the CARDAMOM ensemble, and the observation estimate arises largely because of the large uncertainties providing a weaker constraint on the calibration. Nevertheless, the median simulated C stocks are within the uncertainty bounds of the observations.*

Several arguments in the conclusion section sound a bit misleading to me, e.g., "failure to account for sub-pixel ecosystem heterogeneity within MDF inversions leads to bias in the flux estimates", "stratification improves flux estimates", and "ecological fidelity of the calibrated model parameters is enhanced". The differences in RMSE and bias between baseline and stratified cases are not very significant, and sometimes the biases (absolute values) are even greater in stratified cases (e.g., -584 gCm-2 in baseline Cwood and -627 gCm-2 in stratified Cwood at 0.05deg scale). More validations of model estimates and constrained parameters should be included to draw such conclusions. Regarding "stratification improves flux estimates", the authors might want to say that "stratification reduces flux uncertainties". If yes, however, lines 287-289 already demonstrate that the reduced uncertainty is a result of assuming independence between strata, and uncertainty with full correlation across strata is comparable to the baseline uncertainty. Please clarify these arguments.

*These arguments relate directly to the contrast between emergent scale-variance in the baseline experiments, which do not account for sub-pixel heterogeneity, and the apparent scale-invariance in the stratified experiments.*

*From this we can conclude that in the baseline case, the C balance analysis was systematically biased, and that this bias increased at coarser resolutions, and did not converge at the finer grid resolutions, which are at the limit of what is currently feasible for large-scale MDF applications with full-parameter retrieval. This scale-dependency in the analysis output should not be conflated with CARDAMOM's calibration to its*

*observational constraints, which will emerge as a balance between observation uncertainty and between observation types.*

*Our experiments also show that the emergent differences in ecological function embedded in the model parameters is lost through aggregation to coarser model domains, and that this information loss is exacerbated at coarse resolution, and in model pixels with more heterogeneous land use. The stratification approach largely resolved the scale-dependence issue.*

*However, we note that ecological fidelity of estimated traits, stocks and fluxes is not solely determined by the degree of scale-variance, but also depends on the salient ecological processes being captured by the model structure, and on the quality of the assimilated data, and will ensure that this is emphasised in the revised manuscript:*

*For example, in the conclusions, the specific reference to ecological fidelity has been modified from*

*"(iii) by separately analysing distinct ecosystems within fragmented landscapes, the ecological fidelity of the calibrated model parameters is enhanced, enabling more robust ecological forecasting and raising the prospect of mapping spatial variations of ecosystem functional traits based on a diverse range of EO data."*

*to:*

*"(iii) by separately analysing distinct ecosystems within fragmented landscapes, the loss of ecological information associated with aggregation to coarse resolutions is limited. Where the observations are accurate and model structure appropriate, this this should improve the ecological fidelity of the calibrated models, enable more robust ecological forecasting, and raises the prospect of mapping spatial variations of ecosystem functional traits based on a diverse range of EO data"*

*We do not state that "stratification reduces flux uncertainties" beyond the extent to which the strata are considered independent, and explicitly state that the degree to which the simulated uncertainties are reduced is dependent on this assumption. However, we acknowledge that the statement in the conclusions that "stratification improves flux estimates" would be better-supported with independent validation, as suggested by the reviewer. As a result, we have changed the wording here to reflect the aspect that we feel is well demonstrated by our experiments, notably that after stratification, flux estimates were relatively insensitive to resolution:*

*"stratification reduces the scale-dependence of flux estimates, facilitating scaling of CARDAMOM applications across larger spatial domains"*

*This result facilitates scaling of CARDAMOM applications across larger spatial domains, and where the model structure and available observations provide a reasonable reflection of the ecosystems present, should therefore lead to improved flux estimates in heterogeneous landscapes.*

I was confused about the reason for invariance in biogenic fluxes with respect to both resolution and method. Does this indicate the biogenic processes in DALEC are (almost) linear ecological processes? The linearity in GPP estimation could be possible, but I am not sure if Reco should have a similar pattern.

> *The insensitivity to resolution and method of the biogenic fluxes was an unexpected result. At first order, GPP is a function of leaf area and environmental factors. Autotrophic respiration is estimated based on a fixed fraction of GPP, and heterotrophic respiration estimated based on a first order turnover rate with an exponential temperature sensitivity. We note that mean LAI is preserved across the domain, and that the data underpinning the meteorological forcing is relatively coarse resolution (0.5deg). As a result there are no significant changes in temperature, VPD, radiation etc. across scales which would impact on both rates of photosynthesis and heterotrophic respiration, which combined with meteorological forcing that has limited spatial variation, would lead to biogenic fluxes (GPP, Ra and Rh) that are to scale, or stratification.*

Minor:

Line 123, should be 30,000 km2?

> *Thank you for spotting this. We will correct this in the revised manuscript.*

Line 293, how does DELAC simulate fire effects? Why fire was negligible?

> *Fire was negligible because the MODIS burned area data indicated very little fire activity across the region. Across the entire domain and time period, only X ha of fire activity was detected. In the finest resolution domain, this fire activity was limited to affecting only 16 pixels. Emissions from fire are estimated by assuming that a fraction of simulated biomass in each of the pools either undergoes combustion, or is transferred to the litter pool. The fraction of each pool affected is determined by calibrated parameters relating to combustion completeness and resilience. Given that fire plays such a limited role in this study area, we did not provide a more detailed description of the implementation of fire disturbance within the model, referring instead to an earlier publication (Exbrayat et al., 2018), where the fire model is described in more detail.*

> *We have added the following clause to the sentence to clarify the limited impact of fire in the study area:*

> *"affecting only sixteen pixels across the finest resolution domain across the entire period of analysis"*

**Reply to RC3**

Dear authors,

I feel like this study and methodology of cardamom represents a major advancement in model calibration. It is particularly exciting to see a framework that could run autonomously using earth observation data. The reproducible nature of this data fusion and calibration process, when coupled with the Bayesian methodology for error estimation (and propagation) could provide more iterable forecasts of carbon or other ecological processes. The question of how spatial

heterogeneity plays out in cellular automaton models with single calibrations is crucial to future forecasting and I appreciate the focus on both natural and anthropogenic disturbances. Further, I appreciate the authors' efforts to address the problem of Jenkin's inequity and the role scale selection plays in informing the forecasting and responding. I think the authors' use of stratification by land cover represents a relatively straightforward, logical, and widely available method by which to create more representative models.

However, I feel that some of the conclusions may overreach their results (particularly without independent evaluation). While this methodology has the possibility to improve the estimation of fluxes (etc.), it is not validated to have done so against measurement. The paper presents a gap in understanding to what degree model performance was improved while making some strong claims of the level of ecological fidelity it is able to preserve. I feel this study is both novel and relevant. I would like to see the authors either change the language and address more of the existing limitations of this study or provide further validation to some of the authors' larger claims. I look forward to receiving your response and want to thank you for conducting great work.

*Thank you for your detailed review and overall positive assessment of our manuscript. We acknowledge that our flux estimates do not have independent validation, and thus do not demonstrate that the fluxes after stratification are improved. The primary aim of this paper is to understand the impact of resolution on the simulated C dynamics of heterogeneous landscapes. In this regard we show clearly that if heterogeneity is ignored, there is a systematic scale-dependent bias, which decreases at finer resolutions, although notably didn't converge at our finest spatial resolution. This scale-variance was much less significant after we stratified the landscape into broad land cover classes.*

*In terms of ecological fidelity of the model representation, we argue that scale-variance is problematic. If strong scale-variance is present, the fidelity of the model to ecological processes is degraded because a potentially significant component of the simulated dynamics is determined by the characteristics of the model domain rather than the ecosystem being studied. However, we acknowledge that scale-invariance is not the only contributing factor towards ecological fidelity, which also depends on the model structure adequately representing the important ecological processes, and on the data (and uncertainties) accurately describing the relevant components of the ecosystem and their evolution through time. In our model experiments, our focus was on this issue of scale-variance, thus we used the same model structure irrespective of the land-use stratum. Clearly this means that the fidelity of the model could be improved (assuming there is sufficient data to support the increase in complexity) by (i) including ecosystem-specific models, e.g. to describe crop development and management in arable systems; (ii) adding stratum-specific parameter priors, e.g. leaf lifespan; and (iii) handling stratum-specific data issues, e.g. excessive seasonality of conifer woodlands. Including any of these additional factors in these experiments would, however, compromise the issue of scale-dependence in the comparison between the stratified and unstratified models, which is the key comparison underpinning the motives for this work. Indeed, stratification is pre-requisite to implementation of these additional refinements to model fidelity. Future work will refine the ecological fidelity of the models used, and will compare regionally aggregated fluxes against top-down atmospheric inversion estimates of the biospheric flux to test the extent to which attempts to improve ecological fidelity are able to resolve existing discrepancies between bottom-up and top-down approaches.*

*We will endeavour to improve the discussion of limitations of the analysis in the revised manuscript.*

General comments

The problem of Jenkin's inequity in landscape or earth systems modeling is a valid and underrepresented viewpoint. However, the answer the authors' model provides can not solve this. Raising this in the introduction raises the idea that authors' methods will be solving or improving on the current structure. Please address in the discussion whether authors feel results provide further proof for Jenkin's inequity or whether they work to address it. This paper does not clearly quantify the advantage of picking one scale (sub-degree +LUC ) in a scale variant system. Some may seem likely or self-evident but would need to be proven. For example, the effect of Jenkin's inequity might be quite similar from the cellular to sub-degree+LUC scale when compared to the explicit plant scale. The authors do however do a great job quantifying that there is an amount of scale variance in this model. Quantification and discussion of how this scale variance impacts forecast, when compared to observable phenomena, would provide a lot of support to this paper.

*We agree with the reviewer Jensen's inequity is an important challenge for land surface modelling. It is a challenge for all modelling applications, for which the system must be aggregated to a practical resolution in both space and time. This challenge is particularly valid for large-scale data assimilation efforts, where the scale of the analysis dictates grid resolutions that may typically be orders of magnitude coarser than the grain of the ecological fabric. In this work, we wanted to explore the extent to which C-cycle diagnostics from the CARDAMOM model-data framework was scale-variant with respect to spatial resolution. We show that the strongest scale-dependence was in disturbance fluxes. To a large part, this is driven by the spatial correlations between simulated disturbance processes (tree cover loss) and landcover where C stocks are concentrated (woodlands). In contrast, biogenic fluxes are more evenly distributed across the landscape, and we observe less scale variance. We note that we do not resolve fine-scale meteorology in our finer resolution domains, and that doing so could potential provide another source of scale-dependence, although based on previous work on meteorological uncertainty in data assimilation for C-cycle applications (Spadavecchia et al., 2011), we expect this to be subordinate to the errors arising from the inaccurate parameter representations that arise from failing to discriminate functionally different ecosystems.*

*We argue that our results show that when we aggregate data over coarse pixels prior to assimilation, as is common practise for data assimilation applications, the diagnostics of the C cycle are scale-dependent. Across the range of resolutions we tested, this scale dependence was largely resolved after a broad stratification of the landscape. As the reviewer correctly remarks, we do not fully "solve" Jensen's inequity, but we can say from our experiments that the effects of Jensen's inequity were reduced through stratification. There is information loss when aggregating individual strata, and if we were to extend our analysis to finer spatial resolutions (e.g. individual stands), we may observe some additional scale dependence in the simulated C dynamics. However, moving to such fine scales is impractical for large-scale applications, and where resolving such fine detail is deemed critical, an alternative sampling strategy may be better suited. A good example of such a situation is in the UK-wide assessment of the UK grassland C balance by Myrgiotis et al. (2022). In this study they sampled and assimilated individual fields within each*

*coarse grid cell, specifically because for this application, part of the retrieval included identification of specific management events (cutting and grazing). Sampling presents its own challenges in scalable applications in other ecosystems, such as forests, where disturbance is highly localised, and may be missed by sampling approaches. Developing and comparing these approaches would be an interesting avenue for future work.*

*We hope that are approach and conclusions are now clearer for the reviewer, and will present key elements from the above in the discussion. We have also modified the title to avoid giving the impression that Jensen's inequity is "solved". However we have left the discussion of Jensen's inequity in the introduction because it provides the rationale for undertaking the scale experiments.*

*In terms of the forecasts, we provided a limited analysis as we were conscious that we did not have validation of forecasts even for a short period. There is deviation between the most-likely estimates for the future C balance between the stratified and baseline approaches. The forecast uncertainties are very large, in part because the residence times in the C pool are also uncertain (see Smallman et al., (2021) for an illustration of this), and we are wary to over-interpret. However, the median estimates do show divergence of the individual strata, and this demonstrates the practical significance of stratification when forecasting with coarse-resolution models. These differences are maintained across the spatial resolutions, and we will present additional comparable figures for the other domains in the SI. There is little scale-dependence in the forecasts, but we did not try to extend the disturbance regimes beyond the end of the calibration period. As a result, the forecasts only show the impact of climate on the biogenic fluxes, which our earlier analysis suggested were not strongly sensitive to the range of resolutions explored in our analysis.*

If I understand the authors' methods correctly, for the baseline model the authors parametrize each pixel separately and in the stratified version the authors separately parameterize each pixel and each pixel's land cover. Is there no information shared across pixels or land cover? Given that land covers would presumably share ecological properties, what is the advantage of not using a hierarchical Bayesian, with priors informed by the larger population of land covers or a bayesian mixed model approach? Please either clarify the decision to make this choice or discuss further the limitations of the separate pixel approach.

*We know that traits can vary as much within a given biome as between biomes. Pixel-independent calibration allows functional traits (as expressed by the model parameters) to vary in space. While we do not suggest that our approach resolves Jenkin's inequality, we do suggest that by targeting individual ecosystem types the parameter retrievals should more accurately reflect the ecosystems of interest. Moreover, by increasing the coordination between ecosystem type and observations, this results in the analysis partitioning uncertainty to the appropriate ecosystem (e.g. coniferous vs grasslands), thus allowing subsequent efforts to target the largest uncertainties.*

*In the stratified case, we do not share information across pixels. In part we did not want to make any changes to the assimilation scheme beyond stratification to avoid complicating the interpretation of the comparison against the behaviour of the baseline experiment across spatial resolutions. Instead, we focus on utilising datasets available at high resolution and thus sit within our framework.*

*Of course, stratification open up possibilities to improve the ecological information that we apply to the model, either through improved model structures that better represent specific ecosystems (e.g. crop development), or refining the parameter priors (e.g. leaf lifespan) to represent the ranges found in specific ecosystems.*

*The suggestion to explore hierarchical Bayesian is a very interesting one. One particularly powerful advantage to a hierarchical approach is that it could open up possibilities to assimilate coarser resolution satellite-derived products that describe important components of the terrestrial C cycle (e.g. GPP) or top-down atmospheric inversions (e.g. NBE), which are currently too coarse to relate to individual ecosystems within heterogeneous landscapes. Implementing a hierarchical system is beyond the scope of this work but would be a valuable avenue to develop in future research to maximise the value of the potential observation constraints that current and future satellite missions are likely to provide at a range of spatial resolutions.*

Given that none of your parameters seem to directly map onto disturbance, is your model capturing the heterogeneity on the landscape, or overfitting a model? Perhaps some of my concern comes from a lack of understanding of how harvest or fire operates in this model (see below). Harvest would be inversely correlated with the likelihood of future harvest at certain temporal scales and correlated at others. At the scale of a few decades recovering stands would likely (though not exclusively) experience an increase in GPP as forests regrow. Given the stationarity of your parameterization, how does your parameterization constrain such instances?

*Ecosystem disturbance (whether biomass removal or fire) is imposed on the model based on observational information, rather than a data point for DALEC to be calibrated on. As a result, we do not have to contend with challenges around likelihood of subsequent disturbances.*

*We recognise that our retrieved parameters are time invariant. Consequently, these parameters may not correctly represent time-varying ecosystem properties, such as shifting allocation of photosynthate during post-disturbance recovery. This is an ongoing challenge with the majority of process-orientated models used in major model intercomparisons due to the uncertainty in how ecosystem parameters evolve. Thus, including this aspect is out of scope of the current analysis. However, CARDAMOM and DALEC have previously demonstrated a capacity to be calibrated to and evaluated with independent site level observations for aggrading forests across multiple decades (Smallman et al., 2017).*

Hypothesis three- could be improved. To test that any two methods of model parameterization will have contrasting parameters is almost by definition true. Further, they will always have divergent projections on some level. Please provide better constraints to make this hypothesis falsifiable or use a more stringent definition of the contrasting and divergening of parameters.

*The key component of H3 is the following clause that aggregation will degrade the ecological information embedded in the retrieved parameters. This is important because current comparable methods to data assimilation frequently rely on "pixel-aggregate" parameterisations. The contrasting parameter sets and divergent evolutions follow directly from the different calibrations as the reviewer highlights, and we appreciate that these components would be better communicated as the consequences of the*

*degradation of ecological information, rather than the starting point for our hypothesis. We will rephrase the hypothesis statement to make this clearer.*

*It now reads:*

*"without accounting for fine-scale variations in land cover, the ecological information embedded within the retrieved parameters will be degraded when assimilating data streams at coarser resolutions, resulting in divergent carbon dynamics in simulations of future trajectories."*

Specific comments

Line 155: Please provide either here, in the results, or in the appendix the results of the MCMC process. Or how your criteria for model convergence. Accepted sample rates, plots of autocorrelation, and hyperparameters provided are all necessary to determine if confidence intervals are reasonable.

> *We have clarified the MCMC process in the methods, adding the following sentences: "We use the Gelman-Rubin's convergence criterion to determine whether multiple chains at each pixel have converged. The Adaptive-proposal MCMC (Haario et al., 2001) does not stipulate or target an acceptance rate; the emergent acceptance rate typically varies between 5 and 25 %. The covariance matrix used in adapting the parameter sampling is generated from an initial phase of the MCMC. No hyperparameters are estimated as part of the process. "*
>
> *However, given that our analysis includes several thousand individual pixels, generating per pixel figures would be impractical and digress from the main focus of the paper.*

Line 167: Given the importance of EDCs in determining this you should list them plainly, and discuss the constraint they do or do not provide with regard to the function you are trying to achieve.

See: Buotte, P. C., Koven, C. D., Xu, C., Shuman, J. K., Goulden, M. L., Levis, S., ... & Kueppers, L. M. (2021). Capturing functional strategies and compositional dynamics in vegetation demographic models. Biogeosciences, 18(14), 4473-4490.

> *We will provide a more detailed discussion of the EDCs in a revised manuscript. However, a complete list and description of EDCs is available in cited literature (Bloom et al., 2016). Our updated methods now includes:*
>
> *"EDCs comprise a series of mathematical rules and functions that impose conditions on the inter-relationships between model parameters to ensure ecological "realism" in the accepted parameter sets, based on ecological theory (Bloom and Williams, 2015). For example, turnover of the wood carbon pool must be slower than foliage turnover. Where EDCs are not satisfied, the likelihood is set to zero. By restricting the acceptable parameter space, the EDCs therefore reduce the effective model complexity (Famiglietti et al., 2021), and tend to reduce bias and equifinality in the calibrated ensembles (Bloom and Williams,*

2015). *The resulting ensemble of parameter sets encapsulate the uncertainty in the calibration within the available observational constraints."*

Line 256 (Disturbance): My apologies if I misunderstand this in other comments. Can the authors please provide greater detail on how disturbance is implemented in the model? This paragraph deals primarily with how it is constrained. There are no direct parameters listed in table A1. If I interpret this correctly, did the authors remove a percentage of tree cover or carbon % to match these data sets? Again, given that this is one of the key differences in the stratified model, a better understanding of the disturbances function in the model is important.

> *We will improve the methodological description so that this is clearer for the reader:*
>
> *To convert area estimates of tree cover loss into changes in C stocks, we use a simple clearance model in which a fraction of the C stored in $C_{wood}$, $C_{foliage}$ and $C_{labile}$ is removed based on the pixel fraction (or stratum-specific subpixel fraction) identified within the GFW dataset as experiencing tree cover loss. In practise, most tree cover loss occurs in the conifer woodlands, and is therefore concentrated in these woodlands in the stratified analysis, compared to the baseline experiments, in which we do not consider the sub-pixel distribution of land cover.*
>
> *For the specifics of the disturbance method, please refer to the earlier General Comment on disturbance.*

Line 320: Shredding of information implies a specificity not realized here. Information loss is inherent in all models. Without estimating the level of information that is lost, shredding seems overly evocative.

> *We have softened the language according to the reviewer's suggestion, for example, this particular sentence now reads:*
> *"In the baseline experiments, comparing across domain resolutions, it is apparent that aggregation to coarser spatial resolutions reduces the range of retrieved traits. The reduction in the widths of the retrieved parameter distributions highlights the loss of information relating to variations of fundamental aspects of ecosystem function in the baseline experiments, as the resolution of ecological gradients is lost. In contrast, it is evident that stratification leads to a reduction in the ecological information loss when aggregating to coarser resolutions, as the widths of the aggregated parameter distributions is maintained."*

Line 361: While more consistent, what evidence do we have that the prediction is significantly or functionally different from the baseline model? The confidence intervals seem to overlap significantly.

> *The calibration is made using identical model structure and the same assimilated data, gridded to different domain resolutions. We argue that the differences across spatial resolutions should be regarded as systematic, and therefore the systematic offset of ensemble medians is the key characteristic to be considered.*

Line 376: I feel this sentence speaks to my larger concerns. There is no way to say that disaggregation ensures the ecological fidelity of a system. Ecology is also scale-dependent.

Further, without validation by observation, there is no way to know that the version outperforms the previous version, given that Jenkins inequity would be a property of this scale as well.

*We have made it clearer throughout the manuscript that the ecological fidelity of the model ensemble is not just related to how data are aggregated, but also to the quality of the data, and the degree to which the model structure captures the key ecological processes. Without direct validation, the reviewer is correct that we cannot test the extent to which the changes to the framework lead to improvements in flux estimates. However we can use the relationships between the experiments (i.e. across resolutions, and between stratified and baseline) and make inferences about whether the scale at which data are aggregated systematically impacts on the parameter retrievals and flux estimates. Given that aside from stratification, the model structure and assimilated data sources and pre-processing are the same, these systematic shifts for the baseline experiments reveal the presence of scale-dependent biases, and the contraction of the parameter ranges indicates significant loss of ecological information encoded into the observations. There is undoubtedly information loss too when using stratification in coarser domains, but critically, this information loss does not prevent the retrieval of parameter ensembles that reflect the functional differences between, for example, conifer woodlands and arable land. We have rewritten the paragraph to make these points more clearly, and qualify the extent to which we can draw conclusions around ecological fidelity.*

*"We found that by stratifying the landscape prior to MDF, the variability in ecosystem function exhibited between ecosystems, manifest in their retrieved parameters, was retained across the range of spatial resolutions considered (Figure \ref{fig:retrieved_traits}). Conversely this ecological information is degraded if data are aggregated without considering \textit{a priori} the underlying distribution of land-use and land-cover, demonstrated by the contraction of the distribution of median parameters across the model domains in the baseline experiments compared to the stratified experiments. Critically, this misrepresentation of ecosystem function was exacerbated at the coarser grid resolutions commonly employed in large scale MDF applications, demonstrated by the contraction of the distribution of median parameters across the model domains. The degradation of ecological information is exemplified by our attempts to constrain mean residence times in the long-lived wood and soil pools, which are critical for understanding the potential carbon sink of terrestrial ecosystems* (Luo et al., 2015; Smallman et al., 2021)*: in the baseline experiment, where data were the relationship between stocks and land-use was ignored, the longer residence times specific to woodland ecosystems (MRT$_{Wood}$; Figure \ref{fig:retrieved_traits}) and heathland areas supporting C-rich peat deposits (MRT$_{Soil}$; Figure \ref{fig:residence_times_dom}) were not well-represented by the posterior parameter estimates, particularly in the coarser model domains. In this sense, stratification led to the retention of greater ecological fidelity in the model ensemble when aggregating to coarser spatial resolution domains. The overall ecological fidelity of the model representation will also be limited by the process representation embedded in the model structure, and in the fidelity of the observation data to the relevant characteristics of the actual ecosystem. Prior research has demonstrated that CARDAMOM can retrieve trait differences across biomes* (Bloom et al., 2016; Smallman et al., 2021)*. We demonstrate that CARDAMOM can also retrieve ecosystem-specific traits in mosaic landscapes when the assimilated data is stratified based on prior knowledge of land-use, as far as the observations available for assimilation faithfully convey the ecosystem characteristics. Given the relative importance of*

*parameter uncertainty on future trajectories of the terrestrial C cycle* (Smallman et al., 2021)*, stratification also presents significant opportunities to take advantage of the Bayesian framework embedded within CARDAMOM by taking advantage of the prior information on land-cover and land-use, for example using global trait databases* (e.g. Kattge et al., 2020)*, to inform parameter prior estimates."*

Line 385: Different modeling frameworks providing different (though I would not say divergent) outcomes are highly likely. I feel your argument would be improved if you would better quantify or qualify the significance (either statistical or practical) of this level of difference.

*The practical significance relates to the fact that by stratifying, we not only get different parameters, but that the parameters reflect distinct ecosystems. For many stakeholders, the "community-average" dynamics of a pixel are not particularly informative, and there is much greater potential utility for models that can be related to specific ecosystem types.*

Line 410: Given static and statistical parameterization, it would be nice to understand the climate change implications of the stratification approach.

*The projections represent the evolution of our calibrated model to future climate and $CO_2$. Model parameters do not vary over time, thus we do not account for shifts in species composition that may occur and drive functional shifts in the terrestrial C cycle. Likewise we make no attempt to simulate adaptive management strategies. In terms of the implications for climate change forecasting generally, the static representation of model parameters is likely to become increasingly limited further into the projections, as one might expect that adaptation to the shifting climate become increasingly significant. Furthermore, the lack of process representation for managed landscapes (harvest, grazing) limit the extent to which climate-change impacts should be interpreted based on this analysis. Instead the projections as shown are intended to highlight the different trajectories for the model parameters calibrated with different strata, and therefore the utility of stratification for making the calibrated models informative for a range of stakeholders who might be interested in individual ecosystem types, rather than the aggregated impact of climate change on the range of ecosystems present.*

*We have added:*

*"The static parameterisation also presents a limitation when forecasting ecosystem responses to climate change, as we do not account for either adaptive management strategies, or shifts in species composition that may drive functional shifts in the terrestrial C cycle. Consequently, forecasts become increasingly uncertain as the environmental conditions deviate from the calibration period. Moreover, capturing these complex functional responses to future climate within ecosystem models remains a major challenge* (Fisher and Koven, 2020)*."*

Line 436: The more you stratify a single cell, the greater proportion of it would be captured by this edge or gradient space. If the gradient space has unique ecosystem properties, is there a point where further stratification would further miscalibrate the model?

If helpful, see: Cushman, S. A., Gutzweiler, K., Evans, J. S., & McGarigal, K. (2010). The gradient paradigm: a conceptual and analytical framework for landscape ecology. In Spatial complexity, informatics, and wildlife conservation (pp. 83-108). Springer, Tokyo.

*This is an interesting question, and an interesting paper – thank you for highlighting it. When applying CARDAMOM to a gridded domain, we calibrate all the parameters of the model independently at each pixel. One of the key motivations for this approach is that it enables the retrieved parameters to vary across environmental gradients. However, our ability to resolve these gradients is limited by the resolution of our model domain given the computational expense for calibration. In our stratification approach we therefore resort to discrete categorisation of the landscape within each pixel. For each stratum, each pixel is still calibrated independently to the others, so that, for example the parameterisation for conifer woodlands can exhibit gradient responses across the model domain as indicated by the observations.*

*Specifically relating to edges. We might expect that there are functional shifts close to ecosystem boundaries. This could be tested in future work by including a stratified class to represent edges, although beyond the scope of this paper. However, within this analysis pixels with more fragmentation will provide parameter representations reflecting the impacts of this fragmentation on the C-cycle, within the limitations of the data and model structure.*

*It is not clear exactly what the reviewer is suggesting through the phrase "whether further stratification would further miscalibrate the model". This ecological edge space is present irrespective of the degree of stratification. To defend further stratification, one would need adequate data on which to base the stratification, and of course, as the stratification becomes finer, issues relating to noise in the assimilated data will become increasingly prevalent.*

Line 459: Again, accounting for subcellular processes at the scale you provide by stratification likely also has a high amount of ecological information loss.

*We do not contest that our stratification approach results in significant levels of ecological information loss when aggregating to coarse resolutions. Ecological information loss is inevitable in the approximations required to produce models. What our experiments show is that the model parameters obtained across spatial resolutions are generally consistent for individual strata across resolutions. This is particularly clear when considering mean residence times for the woody/structural C pool (Cwood). In the stratified cases, we see higher residence times for Cwood in the woodland classes than the non-woodland classes, and that these distinctions are preserved across spatial resolutions (Figure 8). In contrast, the baseline case exhibits a contraction in the range residence times to an intermediate value, consistent with the distinct ecosystem characteristics being blended, and resultant parameter values that no longer reflect the actual distinct ecosystems present. To highlight this further, we have added rows to Figures 8, A6, A7 and A8, which show the combined parameter traits expressed across spatial resolutions when aggregating the strata separately. This highlights the difference in information loss between the two approaches.*

*Throughout the manuscript we have made changes to our language to communicate more precisely and make it clear that while some information loss is inevitable when aggregating to coarse resolutions, the degree of information loss is much lower when using stratification. In Section 3.3., we have the following:*

*"The stratified data assimilation scheme reveals emergent differences between ecosystems, while traits retrieved for the baseline experiments characterised intermediate values (Figures …). In the baseline experiments, comparing across domain resolutions, it is apparent that aggregation to coarser spatial resolutions reduces the range of retrieved traits. The reduction in the widths of the retrieved parameter distributions highlights the loss of information relating to variations of fundamental aspects of ecosystem function in the baseline experiments as the resolution of ecological gradients is lost. In contrast, it is evident that stratification leads to a reduction in the extent of ecological information loss when aggregating to coarser resolutions, as the widths of the aggregated parameter distributions maintained."*

*Additionally, in the conclusions, the specific reference to ecological fidelity has been modified from :*

*"(iii) by separately analysing distinct ecosystems within fragmented landscapes, the ecological fidelity of the calibrated model parameters is enhanced, enabling more robust ecological forecasting and raising the prospect of mapping spatial variations of ecosystem functional traits based on a diverse range of EO data."*

*to:*

*"(iii) by separately analysing distinct ecosystems within fragmented landscapes, the loss of ecological information associated with aggregation to coarse resolutions is limited. Where the observations are accurate and model structure appropriate, this this should improve the ecological fidelity of the calibrated models, enable more robust ecological forecasting, and raises the prospect of mapping spatial variations of ecosystem functional traits based on a diverse range of EO data"*

Line 467: While conceptually likely that this provides improved flux estimates, I don't think you have provided enough validation to show this is true. Reduced parameter uncertainty does not dictate estimation capability. Also, if I understand section 3.1 correctly then the parameter uncertainty is roughly similar, though the means may converge indicating some level of reduced scale variance. That this method reduces scale variance, does not directly imply improved estimation.

*We acknowledge that the statement in the conclusions that "stratification improves flux estimates" would be better-supported with independent validation, as suggested by the reviewer. As a result, we have changed the wording here to reflect the aspect that we feel is well demonstrated by our experiments, notably that after stratification, flux estimates were relatively insensitive to resolution:*

*"stratification reduces the scale-dependence of flux estimates"*

*This facilitates scaling of CARDAMOM applications across larger spatial domains, and where the model structure and available observations provide a reasonable reflection of the ecosystems present, should therefore lead to improved flux estimates in heterogeneous landscapes.*

Figure 9: I feel that this figure is crucial to your larger argument of scale-dependent outcomes impacting future projections. I feel several aspects of this figure should be revised. Do these model runs represent the median trait estimation or a single draw of the cardamom traits? Please explain in the text. Further, why is the error not propagated here, given the Bayesian approach? This seems crucial to the case that these methods result in fundamentally different models. It is hard for me to understand the implementation (or lack thereof) of disturbance in these forecasts, given that none of the parameters presented would represent that explicitly. See the above comment, some of this may be a misunderstanding of how disturbance works within the model. If the disturbance is only applied top-down, do these projections represent what you captured (that disturbance is highly scale-relevant)?

*The purpose of Figure 9 is to highlight the different trajectories that might be simulated for the individual strata. This has a practical significance when considering the utility of the derived models, since in many circumstances stakeholders may be interested in specific components of the landscape.*

*We will clarify the figure description in the revised manuscript. Currently plotted are the median estimates from the projected ensemble.*

*The propagated errors are not shown because overlapping uncertainties for six time series are difficult to display in a single plot. We will include supplementary plots to convey this information as clearly as possible. Also in the text we now refer directly to projection uncertainties being large:*
*"… although the projection uncertainties are large (Figure \ref{fig:forecast_summary}), reflecting the significant role of parameter uncertainty in forecasts \citep{smallman2021}."*

*Regarding disturbance, in our model this is imposed top-down based on observations of tree cover loss. We do not attempt to parameterise the underlying mechanisms of disturbance. For the forecasts we simply did not apply any future disturbance. After aggregating the strata, the combined projections for the stratified approach are quite similar to the baseline approach, bearing in mind the expansion of uncertainty further into the forecast. This is consistent with the result that our simulated biogenic fluxes within the calibration period were not very sensitive to resolution and did not differ greatly between the two approaches. As the projections are driven by biogenic fluxes in the absence of further disturbance, they are not being subject to the aspect of the C-cycle that were most strongly scale-dependent in our analysis.*

*We have added the following into the text:*
*"The forecast simulations do not include any impacts of future disturbance, so the evolution of dC$_{bio}$ post-2020 was driven only by biogenic processes, which were not strongly scale-dependent in our experiments."*

*And:*

*"The future trajectories were comparable across the range of spatial resolutions (Figures …)"*

[revised manuscript text omitted]

---

## Author Response (AR2)

**Response to Reviewer #3:**

D. T. Milodowski, T. L. Smallman, and M. Williams

I thank the authors for their revision's, I believe the language is now more appropriate to the findings, with appropriate caveats. I appreciate the additional figures in the supplemental.

*We are happy to hear that the reviewer is satisfied with the changes made during the last revision.*

Line 20: Please clarify it reads "The differences in simulated disturbance fluxes of terrestrial carbon balance that suggest a C sink in the stratified experiment is weaker than in the stratified experiment." Should one of these be the baseline experiment?

*We thank the reviewer for highlighting this error in the text. We have corrected it so that the sentence now makes sense.*

> *"The differences in the simulated disturbance fluxes result in estimates of the terrestrial carbon balance in the stratified experiment that suggest a weaker C sink compared to the baseline experiment."*

Line 125: Per the comment in my previous review. I still don't know how you test the hypothesis that the parameters will be degraded, without a validation or a known true parameter value. In response to some of my prior comments, the logic is circular; that aggregating degrades the ecological value and so we will test if aggregating the data degrades the ecological value of the parameters. Could you use "there will be greater error in parameter estimations when assimilating data streams at coarser resolutions" or that component sections will distinctly differ from their aggregated means (as is the case in this study with the minority land types)?

*We thank the reviewer for highlighting that this hypothesis statement can be improved.*

*We have paired experimental tests across spatial resolutions, one in which we simply average EO signals across larger pixels, and another in which we stratify based on functional variations in land cover at fine resolutions before aggregating. In both cases, there is inevitable information loss associated with aggregation, however, what is clear from the pairwise comparison is that the baseline simulations demonstrate a clear contraction of parameter estimates towards some "average" parameter value. In contrast, the distributions of parameters retrieved in the stratified analysis are much more consistent across spatial resolutions. To the extent that the parameters carry information about ecosystem function (notwithstanding the important potential issues around model structure, imperfect data etc., which provide additional sources of error), we can say that this represents a loss of functional information in the retrieved parameters. The only difference in the two approaches is the way in which the data are aggregated prior to data assimilation. Thus, the systematic shifts observed in the baseline experiments are predominately driven by averaging the signal to across different land cover types as spatial scale increases. The issue then is not simply that the retrieved parameter estimates are erroneous, but rather that a single parameter set cannot provide an appropriate representation of the functional variation. This is an important consideration when interpreting the parameter retrievals of large-scale spatially explicit data assimilation studies, which necessarily have coarse spatial resolution relative to the functional fabric of the underlying landscape. We acknowledge the hypothesis statement could be better, and have endeavoured to improve it as follows:*

*H3: Aggregating data to coarser spatial resolutions results in parameter estimates that increasingly fail to capture functional variations between land cover types, but stratifying the landscape prior to aggregation will reduce this functional information loss.*